# Green Supplier Selection Based on Green Practices Evaluated Using Fuzzy Approaches of TOPSIS and ELECTRE with a Case Study in a Chinese Internet Company

**DOI:** 10.3390/ijerph17093268

**Published:** 2020-05-07

**Authors:** Guohua Qu, Zhijie Zhang, Weihua Qu, Zeshui Xu

**Affiliations:** 1School of Management Science and Engineering, Shanxi University of Finance and Economics, Taiyuan 030006, China; zzj199805@163.com; 2Institute of Management and decision, Shanxi University, Taiyuan 030006, China; quweihua@sxu.edu.cn; 3School of Economics and Management, Shanxi University, Taiyuan 030006, China; 4Business School, Sichuan University, Chengdu 610064, China; xuzeshui@scu.edu.cn

**Keywords:** green supply chain management (GSCM), supplier selection, triangular fuzzy number, fuzzy TOPSIS, ELECTRE

## Abstract

The selection of appropriate green chain suppliers is a very critical decision for effective and efficient green supply chain management in today’s increased awareness and significant environmental pressures from various stakeholders. The aim of this paper is to screen appropriate green chain suppliers based on a framework using fuzzy TOPSIS and ELECTRE for a Chinese internet company. The framework is proposed, grounded on a literature review on green supply chain management practices, after which an empirical analysis is made to be applied an integrated suppliers selection, based on green practices incorporating specifically data collected of the 12 criteria from a set of 12 available suppliers. We use a fuzzy TOPSIS and ELECTRE approach to rank the green chain suppliers, and the results of the proposed framework are compared with the ranks obtained by both the outranking degrees and the incomparability among the actions of fuzzy ELECTRE methodology. Finally, sensitivity analysis was conducted to test the feasibility of the best alternative. The results indicated that the best supplier was alternative 9, and there were four dominant criteria: management support for GSCM, used environmentally friendly materials, followed legal environmental requirements and policies, and reduced the use of harmful substances.

## 1. Introduction

Recently, resource shortages and environmental pollution have become serious problems facing all countries in the world. Faced with the urgent requirements of resource conservation and environmental friendliness, modern enterprise production management needs to focus on how to balance economic benefits and environmental sustainable development [1]. In the early 1980s, the concept of supply chain management (SCM) emerged in the literature, which refers to the management of materials, information flow and logistics activities within and between companies [2]. Over time, SCM has developed in terms of information flow, internal and external relationship networks, and governance of supply networks [3]. At the end of the 20th century, Green Supply Chain Management (GSCM) emerged as a new management model and gradually gained people’s attention, and pursues both economic benefits and environmentally sustainable development [4]. Now it has been favored by research in various fields. Green product design, green supplier evaluation, green production, green packaging and transportation, green marketing and resource recycling are all key links involved in GSCM [1]. In the whole supply chain, the green supplier is located in the upstream, which plays a great role in cost-saving and environmental protection, and can run through all links of the downstream supply chain [1]. The selection of green suppliers can effectively improve the compatibility and environmental performance of the supply chain, which is the core part of GSCM [5].

Some environmental standards need to be emphasized when choosing green suppliers [6]. In the existing studies, some scholars selected suppliers for standards related to environmental practices [7] or standards related to hazardous substances management [8]. Some scholars used social, economic and environmental practices as criteria when selecting suppliers [9]. In addition, there are some studies that use the company’s GSCM reputation as the standard for selecting suppliers [6]. However, according to the author’s review of the existing literature, it is not found that the relevant research criteria include after-sales service. In the decision-making process of selecting green suppliers, it is necessary to focus on uncertain language and incomplete information environment issues. Sun et al. [10] proposed a weights-determining method in MCDM based on negation of probability distribution. This method combines probability distribution negation with evidence theory to reduce the uncertainty caused by human subjective factors through quantitative evaluation of criterion ambiguity. Instead of being provided in advance by the decision maker, Fei and Deng [11] proposed a new criterion weight determination method based on similarity measures and aggregation operators of PFNs and IVPFNs, which effectively reduces people’s subjective initiative. Therefore, some measures should be taken to improve the reliability of decision results in uncertain environments. 

At the same time, scholars used different research methods for green supplier selection in green supply chain management practices. For example, the extended AHP (FAHP, FEAHP), the analytic network process (ANP), applying chi-square tests to explore operational green supplier performance (Vachon and Klassen (2005)), fuzzy TOPSIS and other individual methodology approaches. There are some integrated methodology approaches such as the aggregated ANP and DEA, the aggregated AHP and DEA and the aggregated ANN, MADA, DEA and ANP. In this article, the fuzzy TOPSIS method is used to study supplier selection in green supply chains. For the study of supplier selection, AHP has a large scale and requires experts to evaluate the attributes in pairs, which is too subjective. In addition, its eigenvalue and eigenvector exact method is quite complex. ANN is too complex, the technical requirements are high, it needs a lot of experience and data support, and is not convenient. DEA is objective, but the number of evaluation indexes is required to be less than the number of suppliers, which is obviously not applicable. The traditional TOPSIS approach is easy to understand and fast, but it does not reflect the preferences of decision makers well. However, the fuzzy TOPSIS method quoted in this paper is convenient and efficient, it will not simply use the traditional fuzzy algorithm and will not lead to a degree of distortion or range too large.

Based on the above analysis, this paper throws away the redundant and similar criteria and proposes 12 criteria that are suitable for the current selection of green suppliers. In addition, a framework of selecting the best green suppliers based on the fuzzy TOPSIS method and the ELECTRE method is proposed, which provides a more reliable practical method for enterprises to choose green suppliers.

The remainder of this paper is arranged as follows. Section 2 sets up some basic background related to green supply chain management, fuzzy TOPSIS and the ELECTRE method. Then, we introduce some theoretical knowledge about fuzzy TOPSIS and fuzzy ELECTRE I in Section 3, which lays a foundation for this paper to study the selection of green suppliers. In Section 4, the method is applied to the supplier selection of a China electronics factory. In addition, a comprehensive sensitivity analysis is performed on the results, calculated using the TOPSIS method to further study the impact of the threshold on the final assessment. In addition, in order to verify the accuracy of the application results of fuzzy TOPSIS method, this paper uses ELECTRE I method to analyze the case, and the final results are similar to the fuzzy TOPSIS technology. Finally, the paper summarizes and analyzes the limitations of this paper and future research direction in Section 5.

## 2. Literature Review

### 2.1. Green Supply Chain Management

In the study of modern operations management, green supply chain management (GSCM) has become one of the most important issues, which is to achieve better environmental efficiency in the supply chain [6]. In fact, green supply chain management is not a concept that all researchers agree on [3]. However, most scholars agree that supply chain management is crucial to an organization’s competitiveness [12]. Government organizations are also encouraging manufacturers to adopt green technologies in different ways, such as subsidizing consumers, subsidizing manufacturers, which has far-reaching implications for the development of sustainable products, manufacturing and remanufacturing decisions [13,14,15]. Additionally, many companies have realized the importance of incorporating environmental protection measures into their daily activities, due to increased awareness of environmental protection and the environmental pressures from various stakeholders [6]. 

GSCM mainly focuses on the improvement of environmental and economic performance [16]. The research shows that GSCM can achieve win-win results in environmental performance and economic benefits [17]. In general, GSCM has a structural relationship between pressure, practice, and performance [18]. We searched the SCI Web of Science database with the keywords “green supply chain” and “green supply chain management”, and produced the 10 most cited works in this field of knowledge. Table 1 summarizes these latest and most influential GSCM works.

We refer to some previous definitions of GSCM:It is an important new model that reduces its environmental risks and impacts, aiming to achieve profit and market share for the enterprise, while improving its ecological efficiency [4].Combine rational environmental management choices with a decision-making process to turn resources into usable products [19].A way for an enterprise to achieve its profit and market share goals by reducing environmental impact and improving ecological efficiency [20].

Many researchers have studied green supply chain practices through various methods [21,22,23]. Kannan et al. [6] proposed a Fuzzy TOPSIS framework for selecting green suppliers for a Brazilian electronics company based on GSCM practice standards. Ageron et al. [12] established a theoretical framework to study through empirical research using the views and practices of selected French companies. By proposing a fuzzy multi-criteria approach, Govindan et al. [9] discussed the problem of determining an effective model for supplier selection operations in a supply chain based on the triple bottom line (economic, environmental, and social) approach. Sarkis and Dhavale [24] proposed a supplier ranking and selection method based on a Bayesian framework and Markov Chain Monte Carlo simulation. Liu et al. [16] first explicitly explored the link between green supply chain management and environmental management from a theoretical perspective.

After reviewing the previous literature, this paper applied the fuzzy TOPSIS method in the study of supply chain management practice, taking an electronics factory in China as an example to conduct corresponding research and analysis on the selection of green suppliers.

### 2.2. Fuzzy TOPSIS

The TOPSIS (Technique for Order Preference by Similarity to an Ideal Solution) method is a method of ranking based on the closeness of a limited number of evaluation objects to an idealized goal, in order to allow us to evaluate the relative merits of the existing choices. However, it cannot avoid the bias caused by vagueness and ambiguity in the decision-making process. Fuzzy set theory is used to solve ambiguity and uncertainty in the decision-making process [25]. Therefore, the combination of fuzzy sets and tendencies will be very suitable for solving group decision-making problems with ambiguity [26]. In recent years, Ghorbani et al. [27,28,29,30] used Fuzzy TOPSIS in their research of supplier selection. Hwang [31] first proposed the crisp TOPSIS method, whose principle is to select the alternative with the shortest distance from the positive-ideal solution (PI) and the longest distance from the negative-ideal solution (NIS) as the optimal scheme, which is a practical method for solving multi-criteria decision-making problems. In recent decades, TOPSIS has been extended to solve MCDM problems in different fuzzy environments [32] such as general fuzzy TOPSIS [33], Extended TOPSIS method based on interval intuitionistic fuzzy numbers [34], IF-TOPSIS [35], hesitant fuzzy linguistic TOPSIS [36], Neutrosophic uncertain linguistic TOPSIS [37] and Pythagorean fuzzy TOPSIS [38].

Fuzzy TOPSIS has been widely used by researchers. Lee [39] applied fuzzy TOPSIS to robust spatial vulnerability or decision-making in water resources management. Samanlioglu et al. [40] combined fuzzy TOPSIS in researching the selection process of the information technology (IT) department of a Turkish dairy company. Krohling and Campanharo [41] proposed a fuzzy TOPSIS method and used it to evaluate the level of the emergency response plan for the simulated oil spill accident. Many scholars use fuzzy TOPSIS when selecting suppliers in the supply chain, such as Kannan [6], Sultana [30], Ahmad Jafarian [9], Jain [29] etc. Through the study of scholars and the development of the TOPSIS method, it can be seen that the fuzzy TOPSIS method is more widely applied and more reliable than the traditional TOPSIS method.

### 2.3. The ELECTRE Method

ELECTRE means elimination and selection in French [42]. Among the MCDA (Multiple Criteria Decision Analysis) methods, the ELECTRE method and its derivatives are among the most popular. The ELECTRE method was originally proposed by Roy [43] and later named ELECTRE I. Then, scholars put forward several other methods one after another: ELECTRE II [44], ELECTRE III [45], ELECTRE IV [46] and ELECTRE TRI [47].

When modeling some imperfect knowledge due to data uncertainty and inaccuracy, outranking methods can consider indifference and preference thresholds. The ELECTRE method and its derivatives play a prominent role in the outranking method. However, each ELECTRE version differs in operation and in the types of problems it can solve. ELECTRE I is used to construct partial priorities, and its goal is to select a set of best alternatives. The task of ELECTRE II, III and IV is to build the order of alternatives from the best to the worst, that is, ranking the alternatives. The ELECTRE III method is more complicated and difficult to explain because this method establishes a outranking degree, which represents the outranking reliability between the two alternatives. ELECTRE TRI is used for problematic sorting, and its purpose is to specify alternatives for a set of predefined categories. The ELECTRE method is widely used in many fields, such as offshore wind power station site selection [48], supplier selection [49], solar photovoltaic farm site selection [50] and performance assessment [51], among others.

The ELECTRE method requires accurate measurement of performance levels and standard weights. However, most decision makers use linguistic terms such as low, medium and high when expressing their judgments, resulting in ratings and weights that cannot be accurately measured in many real world problems [25]. Therefore, the fuzzy outranking method can solve the problem of inaccurate measurement of performance levels and standard weights. Subsequently, scholars introduced fuzzy concordance and fuzzy discordance in the ELECTRE method [52]. Scholars continue to propose innovative methods, and make the ELECTRE method based on vague and uncertain language environment better applied. 

## 3. Solution Methodology

Based on the above research, this paper proposes a framework that companies can use when choosing green suppliers. The framework was determined through literature research and discussion, and adopted a qualitative and quantitative method to select green suppliers, as shown in Figure 1. In addition, this section briefly introduces the solution methodology in the framework. Specifically, fuzzy sets, linguistic variables, fuzzy TOPSIS methods, and ELECTRE I.

### 3.1. Preliminary Definitions

In reality, the natural language of human expression perception or judgment carries subjectivity, uncertainty, and ambiguity. Precise numbers cannot truly simulate the real world, nor can we accurately calculate the importance of objects. In order to solve this kind of problem and deal with the ambiguity caused by the uncertainty and ambiguity in language evaluation in the decision-making process, Zadeh [25] introduced fuzzy set theory to analyze the decision-making problem to express the language terms in the decision-making process. Fuzzy sets can be used to deal with the fuzziness in the subjective judgment of experts [53]. This method of introducing fuzzy set theory into the analysis of decision-making problems is called fuzzy multi-criteria decision-making (FMCDM) [54,55]. In this paper, triangular fuzzy numbers are used to evaluate the decision makers’ preferences in order to facilitate the use and calculation of DMs. Some related definitions of the fuzzy set theory used in this paper are as follows:

**Definition 1.** **(fuzzy set)**
*A fuzzy set is a collection of objects that have the attributes described by a fuzzy concept. Let A be a fuzzy set; $A=\left\{\left(x_{1},m_{A}\left(x_{1}\right)\right)\left(x_{2},m_{A}\left(x_{2}\right)\right)\left(x_{3},m_{A}\left(x_{3}\right)\right)L\left(x_{n},m_{A}\left(x_{n}\right)\right)\right\}$, μ_A_(x) is the membership of element x to fuzzy set A which maps each element x in X to a real number [0, 1] [56].*


**Definition 2.** **(fuzzy number)**
*A triplet (a, b, c) can be used to represent a triangular fuzzy number [56]. The membership function of the fuzzy number μ_A_(x) is given by*
(1)$$m_{A}(x)\{\begin{array}{l} 0,x<a,x>c\\ \frac{x-a}{b-a},a\leq x\leq b\\ \frac{c-x}{c-b},b\leq x\leq c \end{array}$$

*The fuzzy number has been illustrated in Figure 2.*


**Definition** **3**[56]**.**
*Given two fuzzy triangular numbers $A=\left(a_{0},b_{0},c_{0}\right)$ and $B=\left(a_{1},b_{1},c_{1}\right)$, the main operations are shown below:*
(2)$$A\left(+\right)B=\left(a_{0},c_{0},b_{0}\right)\left(+\right)\left(a_{1},c_{1},b_{1}\right)=\left(a_{0}+a_{1},b_{0}+b_{1},c_{0}+c_{1}\right)$$
(3)$$A\left(-\right)B=\left(a_{0},c_{0},b_{0}\right)\left(-\right)\left(a_{1},c_{1},b_{1}\right)=\left(a_{0}-a_{1},b_{0}-b_{1},c_{0}-c_{1}\right)$$
(4)$$A\left(\times \right)B=\left(a_{0},c_{0},b_{0}\right)\left(\times \right)\left(a_{1},c_{1},b_{1}\right)=\left(a_{0}a_{1},b_{0}b_{1},c_{0}c_{1}\right)$$
(5)$$A\left(÷\right)B=\left(a_{0},c_{0},b_{0}\right)\left(÷\right)\left(a_{1},c_{1},b_{1}\right)=\left(\frac{a_{0}}{a_{1}},\frac{b_{0}}{b_{1}},\frac{c_{0}}{c_{1}}\right)$$
(6)$$nA=\left(na_{0},nb_{0},nc_{0}\right)$$
(7)$$A^{-1}=\left(\frac{1}{a_{0}},\frac{1}{b_{0}},\frac{1}{c_{0}}\right)$$

**Definition** **4.**
*Given two fuzzy triangular numbers $A=\left(a_{0},b_{0},c_{0}\right)$ and $B=\left(a_{1},b_{1},c_{1}\right)$, the distance between the two fuzzy numbers is calculated as [56]:*
(8)$$d\left(A,B\right)=\sqrt{1/3\left[{\left(a_{0}-a_{1}\right)}^{2}+{\left(b_{0}-b_{1}\right)}^{2}+{\left(c_{0}-c_{1}\right)}^{2}\right]}$$


**Definition** **5.**
*Suppose that the decision-making committee consists of K decision makers. The positive triangular fuzzy numbers W_k_ (k = 1, 2, …, K) with membership functions μ_WK_(x) (k = 1, 2, …, K) represent the fuzzy ratings of the DM_k_ (k = 1, 2, …, K). Then, the aggregated fuzzy rating can be indicated as:*
(9)$$W_{k}=\left(a,b,c\right),\;k=\left(1,2,\ldots ,K\right)$$
*where*
$a=min_{k}\left\{a_{k}\right\},\;b=1/k\sum_{k=1}^{k}b_{k,c}=max_{k}\left\{c_{k}\right\}$


**Definition** **6.**
*For any fuzzy numbers A and B, the Hamming distance [57] d(A, B) is defined by the formula:*
(10)$$d\left(A,B\right)=\int_{R}\left|\mu_{A}\left(x\right)-\mu_{B}\left(x\right)\right|dx$$
*where R is the set of real numbers.*


### 3.2. Fuzzy TOPSIS

As mentioned before, the fuzzy TOPSIS method is a method that combines the fuzzy set theory with the traditional TOPSIS method. In this method, the evaluation basis of the optimal solution is that the geometric distance between the solution and the positive-ideal solution (PIS) is the shortest under the benefit standard, and the geometric distance between the solution and the negative-ideal solution (NIS) is the longest under the cost standard [31]. The steps of the fuzzy TOPSIS method are as follows:

**Step 1.** Construct the fuzzy decision matrix.

In MCDM problems, there are usually *k* decision makers evaluating *m* choices with *n* criteria. It can be expressed in a matrix as:
$$R=\left[\begin{array}{cccc} r_{11} & r_{12} & \cdots & r_{1n}\\ r_{21} & r_{22} & \cdots & r_{2n}\\ \vdots & \vdots & \ddots & \vdots \\ r_{n1} & r_{n2} & \cdots & r_{nn} \end{array}\right],\;i=1,2,3,\ldots ,\;j=1,2,3\ldots$$

**Step 2.** Normalize the fuzzy decision matrix.

The normalized fuzzy-decision matrix is calculated as:
$$R={\left[r_{ij}\right]}_{m*n}$$

The normalized values for benefit and cost related criteria are calculated as:(11)$$\mathrm{Benefit}\;\mathrm{criteria}:\;r_{ij}=(\frac{a_{ij}}{c_{j}^{*}},\frac{b_{ij}}{c_{j}^{*}},\frac{c_{ij}}{c_{j}^{*}}),\;i=1,2,3,\ldots ,\;j=1,2,3\ldots$$
(12)$$\mathrm{Cos}t\;\mathrm{criteria}:\;r_{ij}=(\frac{a_{j}^{-}}{c_{ij}},\frac{a_{j}^{-}}{b_{ij}},\frac{a_{j}^{-}}{a_{ij}}),\;i=1,2,3,\ldots ,\;j=1,2,3\ldots$$
where $c_{j}^{*}=max_{i}c_{ij}$, $c_{j}^{-}=min_{i}c_{ij}$

**Step 3.** Construct the weighted normalized fuzzy decision matrix.

The weighted normalized value *v_ij_* can be given by:(13)$$V={\left[v_{ij}\right]}_{m*n},\;i=1,2,\ldots ,m;\;j=1,2,\ldots ,n$$
where $v_{ij}=r_{ij}*w_{j}$ and *w_j_* is the weight of the jth attribute.

**Step 4.** Calculate the fuzzy positive-ideal solution (FPIS, A*) and the fuzzy negative-ideal solution (FNIS, A^−^).
(14)$$A^{*}=\left(v_{1}^{*},v_{2}^{*},\ldots ,v_{n}^{*}\right)\;\mathrm{where}\;v_{j}^{*}=max_{i}\left\{v_{ij}\right\},\;i=1,2,\ldots ,m;\;j=1,2,\ldots ,n$$
(15)$$A^{-}=\left(v_{1}^{-},v_{2}^{-},\ldots ,v_{n}^{-}\right)\mathrm{where}\;v_{j}^{-}=min_{i}\left\{v_{ij}\right\},\;i=1,2,\ldots ,m;\;j=1,2,\ldots ,n$$

**Step 5.** Calculate the distance of each alternative from FPIS and FNIS. The distance of each weighted alternative from the FPIS and the FNIS can be represented as:(16)$$d_{i}^{*}=\sum_{j=1}^{n}d_{v}\left(v_{ij},v_{j}^{*}\right)$$
(17)$$d_{i}^{-}=\sum_{j=1}^{n}d_{v}\left(v_{ij},v_{j}^{-}\right)$$
where *d_v_(a,b)* is the distance between two fuzzy numbers, and it is calculated by Equation (7).

**Step 6.** Calculate the closeness coefficient (*CC_i_*).
(18)$$cc_{i}=\frac{d_{i}^{-}}{d_{i}^{*}+d_{i}^{-}},\;i=1,2,\ldots ,m$$

**Step 7.** Rank the preference order. The best alternative is closest to the FPIS and farthest from the FNIS. The alternatives are ranked according to the *CC_i_* in decreasing order. 

### 3.3. Fuzzy ELECTRE I

In this section, we will introduce a hybrid approach that combines the fuzzy set theory with ELECTRE I. The steps of fuzzy ELECTRE I method are as follows [57]:

**Steps 1–3** are the same as **steps 1–3** of fuzzy TOPSIS.

**Step 4.** Calculate the distance between the alternatives A_g_ and A_f_ related to the criterion j using the Hamming distance method.
$$d\left(A,B\right)=\int_{R}\left|\mu_{A}\left(x\right)-\mu_{B}\left(x\right)\right|dx$$

**Step 5.** Construct the concordance matrix (C).
$$C=\left[\begin{array}{cccccc} - & \cdots & C_{1f} & \cdots & C_{1\left(m-1\right)} & C_{1m}\\ \vdots & \ddots & \vdots & \ddots & \vdots & \vdots \\ C_{g1} & \cdots & C_{gf} & \cdots & C_{g\left(m-1\right)} & C_{gm}\\ \vdots & \ddots & \vdots & \ddots & \vdots & \vdots \\ C_{m1} & \cdots & C_{mf} & \cdots & C_{m\left(m-1\right)} & - \end{array}\right]$$
(19)$$\mathrm{where}\;C_{gf}=\left(C_{gf}^{l},C_{gf}^{m},C_{gf}^{u}\right)=\sum_{j\in J_{c}}W_{j}=\left(\sum_{j\in J_{c}}w_{j}^{l},\sum_{j\in J_{c}}w_{j}^{m},\sum_{j\in J_{c}}w_{j}^{u}\right)$$

**Step 6.** Construct the discordance matrix (D).$$D=\left[\begin{array}{cccccc} - & \cdots & d_{1f} & \cdots & d_{1\left(m-1\right)} & d_{1m}\\ \vdots & \ddots & \vdots & \ddots & \vdots & \vdots \\ d_{g1} & \cdots & d_{gf} & \cdots & d_{g\left(m-1\right)} & d_{gm}\\ \vdots & \ddots & \vdots & \ddots & \vdots & \vdots \\ d_{m1} & \cdots & d_{mf} & \cdots & d_{m\left(m-1\right)} & - \end{array}\right]$$
(20)$$\mathrm{where}\;d_{gf}=\frac{\max_{j\in J_{D}}\left|v_{gj}-v_{fj}\right|}{\max_{j}\left|v_{gj}-v_{fj}\right|}=\frac{\max_{j\in J_{D}}\left|d\left(\max\left(v_{gj},v_{fj}\right),v_{fj}\right)\right|}{\max_{j}\left|d\left(\max\left(v_{gj},v_{fj}\right),v_{fj}\right)\right|}$$

**Step 7.** Identify the Boolean matrices B and H based on the minimum concordance level $\tilde{C}$ and the minimum discordance level $\tilde{D}$, respectively.
(21)$$B=\left[\begin{array}{cccccc} - & \cdots & b_{1f} & \cdots & b_{1\left(m-1\right)} & b_{1m}\\ \vdots & \ddots & \vdots & \ddots & \vdots & \vdots \\ b_{g1} & \cdots & b_{gf} & \cdots & b_{g\left(m-1\right)} & b_{gm}\\ \vdots & \ddots & \vdots & \ddots & \vdots & \vdots \\ b_{m1} & \cdots & b_{mf} & \cdots & b_{m\left(m-1\right)} & - \end{array}\right]\mathrm{where}\{\begin{array}{c} C_{gf}\geq \tilde{C},then,b_{gf}=1\\ C_{gf}<\tilde{C},then,b_{gf}=0 \end{array}$$
(22)$$\mathrm{where}\;\tilde{C}=\left(C^{l},C^{q},C^{u}\right),\;\mathrm{and}C^{l}=\frac{\sum_{f=1}^{m}\sum_{g=1}^{m}c_{gf}^{l}}{m\left(m-1\right)},\;C^{q}=\frac{\sum_{f=1}^{m}\sum_{g=1}^{m}c_{gf}^{q}}{m\left(m-1\right)},\;C^{u}=\frac{\sum_{f=1}^{m}\sum_{g=1}^{m}c_{gf}^{u}}{m\left(m-1\right)}.\;H=\left[\begin{array}{cccccc} - & \cdots & h_{1f} & \cdots & h_{1\left(m-1\right)} & h_{1m}\\ \vdots & \ddots & \vdots & \ddots & \vdots & \vdots \\ h_{g1} & \cdots & h_{gf} & \cdots & h_{g\left(m-1\right)} & h_{gm}\\ \vdots & \ddots & \vdots & \ddots & \vdots & \vdots \\ h_{m1} & \cdots & h_{mf} & \cdots & h_{m\left(m-1\right)} & - \end{array}\right]\;\mathrm{where}\{\begin{array}{c} d_{gf}\geq \tilde{D},then,b_{gf}=0\\ d_{gf}<\tilde{D},then,b_{gf}=1 \end{array}$$
where $\tilde{D}=\frac{\sum_{f=1}^{m}\sum_{g=1}^{m}d_{gf}}{m\left(m-1\right)}$. 

**Step 8.** Construct the global matrix Z.
(23)Z=B×H,
where $z_{gf}=b_{gf}h_{gf}$

**Step 9.** Rank the alternatives.

**Step 10.** Depict a decision graph. Figure 3 illustrates the graphical representation of the binary relations (>, =, ?).

## 4. Application of the Proposed Green Supplier Selection Framework

An enterprise’s environmental social responsibility is the responsibility consciousness that every enterprise should have. With the enhancement of people’s awareness of ecological protection, the concept of sustainable development is deeply rooted in people’s hearts. Various domestic policies encourage enterprises to develop in a green way, and green environmental protection is gradually incorporated into the corporate culture of each enterprise. We will demonstrate our application of this method in the selection of green suppliers for a Chinese electronics company. In order to achieve sustainable development, the case company is responsible for the products it sells and the environmental impact it causes. The case company realizes that providing greener electronic products not only better meets the needs of customers but also reflects the social value of the company. Finding good green suppliers is the key to improve its supply chain management. The case company’s sustainability manager seeks a way to identify and select suppliers that support the case company’s adoption of GSCM practices.

In this case, ten major suppliers have been identified as candidates for the company (supplier A1, A2, A3, A4, A5, A6, A7, A8, A9 and A10). According to the principles of scientificity, feasibility, systematicness, independence, hierarchy, comparability, dynamics, flexibility and completeness, this paper designs 12 green supplier evaluation criteria. From the previous procurement and green procurement strategy assessment, and by referring to the supplier selection decision criteria in previous studies, we included 11 of the 17 criteria confirmed by Zhu (2008), deleted redundant criteria and combined similar criteria. During the process of selecting criteria, in addition to the traditional decision criteria for supplier selection, in the process of understanding the company’s actual operating conditions and requirements on its suppliers, we learned that the raw materials or parts provided by previous suppliers did not have good after-sales service, and the company often had to deal with defective products by itself. Through interviews with some other electronic companies, we found a similar situation. In order to realize the sustainable development of the company, a criterion of after-sales service was added after the discussion of experts. The final list contains 12 criteria. These criteria are management support for supply chain management (C1), follow legal environmental requirements and policies (C2), passed ISO 14001 certification (C3), use environmentally friendly technologies and equipment (C4), developing ecological products (C5), use environmentally friendly materials (C6), reduce the use of harmful substances (C7), lean management (C8), sustainable recycling design (C9), reasonable inventory management (C10), conduct internal environmental management evaluation of suppliers (C11), and quality after-sales service (C12). Table 2 presents the details of these criteria.

According to the purpose of the company, we prepared questionnaires to evaluate 10 suppliers, sent the questionnaire to the academic expert group for content analysis and improvement and chose three experienced senior management personnel of the panel (the selected decision makers not only have a deep understanding of the company’s business strategy and operation strategy, but also are familiar with the company’s procurement strategy, supplier selection process and procurement performance results). In this paper, we use 0-10 scale and 0-1 scale to score the alternatives and criteria, as shown in Table 3 and Table 4, respectively. To obtain the weight preference of the criteria, three decision makers (D1, D2, D3) were asked to score the selected criteria according to the language terms in Table 4. Table 5 shows the fuzzy weight of each decision maker on each criterion. According to the language terms in Table 3, each decision maker scores 10 alternatives according to the 12 decision criteria developed. Table 6, Table 7 and Table 8 are the fuzzy decision matrices of three decision makers.

### 4.1. Application of Fuzzy TOPSIS

In the following, we will take the criterion 1 and the alternative 1 as an example to demonstrate the application of the fuzzy TOPSIS method in the proposed framework: 

Firstly, we use Equation (9) to calculate the fuzzy aggregated weights (*w*) of each criterion as follows:
$$a=\min_{k}\left\{0.8,0.8,0.8\right\},\;b=\frac{1}{3}\sum_{k=1}^{3}b_{k}=\frac{1}{3}\times \left(0.9+0.9+0.9\right),\;c=\max_{k}\left\{1,1,1\right\}$$
w={0.8,0.9,1}

Likewise, the remaining criteria can be computed and the result is presented in Table 9.

Secondly, we also use Equation (9) to calculate the fuzzy aggregated weights (*w*) of each alternatives. The rating for A1 for C1 can be computed as:
$$a=\min_{k}\left\{9,7,9\right\},\;b=\frac{1}{3}\sum_{k=1}^{3}b_{k}=\frac{1}{3}\times \left(10+9+10\right),\;c=\max_{k}\left\{10,10,10\right\}$$
w={7,9.667,10}

Similarly, the remaining alternatives can be computed and the fuzzy aggregated decision matrix is presented in Table 10.

Thirdly, we use Equations (11) and (12) to normalize the fuzzy aggregated decision matrix. The normalized rating for A1 for C1 can be calculated as
$$c_{j}^{*}=\max_{i}\left(10,10,10,10,10,10,5,10,10,9\right)=10$$
$$r_{ij}=(\frac{7}{10},\frac{9.667}{10},\frac{10}{10})=\left(0.7,0.967,1\right)$$

We only calculate $c_{j}^{*}$ because all the selected criteria as shown in Table 11 are benefit criteria. The normalized fuzzy aggregated decision matrix is presented in Table 12.

Next, we use Equation (13) to compute the fuzzy weighted decision matrix. For A1, the fuzzy weight of C1 is given by
$$v_{ij}=\left(0.7,0.967,1\right)*\left(0.8,0.9,1\right)=\left(0.56,0.870,1\right)$$

Similarly, the remaining can be computed and the result is presented in Table 13. Next, the fuzzy positive-ideal solution (*A**) and the fuzzy negative-ideal solution (*A^−^*) of the 10 alternatives can be calculated using Equations (14) and (15) respectively.

Next, we compute the distance of each alternative from the positive-ideal solution $d_{v}\left(A_{1},A^{*}\right)$ and the distance of each alternative from the negative-ideal solution $d_{v}\left(A_{1},A^{-}\right)$ using Equation (8). Therefore, for A1 and C1, the distance $d_{v}\left(A_{1},A^{*}\right)$ and $d_{v}\left(A_{1},A^{-}\right)$ can be given by
$$d_{v}\left(A_{1},A^{*}\right)=\sqrt{\frac{1}{3}\left[{\left(0.56-1\right)}^{2}+{\left(0.87-1\right)}^{2}+{\left(1-1\right)}^{2}\right]}=0.2649$$
$$d_{v}\left(A_{1},A^{-}\right)=\sqrt{\frac{1}{3}\left[{\left(0.56-0\right)}^{2}+{\left(0.87-0\right)}^{2}+{\left(1-0\right)}^{2}\right]}=0.8308$$

Similarly, we get the distances of the remaining criteria for the 10 alternatives. The results are presented in Table 14 and Table 15. 

Next, we use Equations (16) and (17) to calculate the distance $d_{i}^{*}$ and $d_{i}^{-}$. For A1 and C1, the distances are given by
$$d_{i}^{*}=\sum_{j=1}^{n}d_{v}\left(v_{ij},v_{j}^{*}\right)=6.479$$
$$d_{i}^{-}=\sum_{j=1}^{n}d_{v}\left(v_{ij},v_{j}^{-}\right)=7.493$$

Similarly, the distances of remaining alternatives from *A^*^* to *A^-^* can be computed. Then, we calculate the closeness coefficient (*CC_i_*) of each alternative. Therefore, the *CC_i_* of A1 is given by
$$cc_{i}=\frac{d_{i}^{-}}{d_{i}^{*}+d_{i}^{-}}=\frac{7.493}{7.493+6.479}=0.536$$

Similarly, the *CC_i_* of other alternatives can be computed. The results are shown in Table 16.

Finally, we compare the *CC_i_* of each alternatives and rank them.

### 4.2. Results

The final results of fuzzy TOPSIS analysis are shown in Table 17. According to the calculation of the closeness coefficient, the ranking of 10 suppliers is:

A9 > A2 > A3 > A5 > A1 > A8 > A7 > A6 > A4 > A10

Based on the ranking, we can say that the supplier 9 is the best selection.

### 4.3. Sensitivity Analysis

The purpose of sensitivity analysis is to examine the influence of choosing different weights for criteria on the selection of green supplier. To perform the sensitivity analysis, we ranked the 12 criteria affecting GSCM by their importance according to the weights given above, as shown in Table 18. In this sensitivity analysis, we conducted 17 cases and their details are presented in Table 19. In the first five cases of sensitivity analysis, we let all criteria’s weights vary from very low (VL) to very high (VH) which are (0,0.2,0.4), (0.2,0.4,0.6), (0.4,0.6,0.8), (0.6,0.8,1), and (0.8,0.9,1). In the 6th case, we set the weight of the first criterion to very high (0.8,0.9,1) and the rest to very low (0,0.2,0.4). In the 7th cases, we set the weight of the 2nd criterion to very high (0.8,0.9,1) and the rest to very low (0,0.2,0.4), and the rest of the cases are done in the same way. We calculate the closeness coefficient of each alternative in different cases, and the alternatives were ranked under 17 different cases. The results of the sensitivity analysis are presented in Table 20 and Figure 3.

According to the sensitivity analysis results in Table 20 and Figure 4, we can see that even if the rank of the green supplier changed slightly with the weight changed, generally, supplier 9 is the best. Supplier 9 has the highest closeness coefficient in 13 cases (case numbers 1–4, 6–7, 9–11, 14–17). Supplier 2 is the best in the remaining four cases. Therefore, it is clear that our decision-making process is relatively insensitive to changes in the weights of the criteria.

### 4.4. Application of Fuzzy ELECTRE I

In this section, we illustrate the application of the fuzzy ELECTRE I method in the framework of this paper:

Firstly, we use Equation (10) to calculate the distance between Ag and Af related to the j criterion, and the results are shown in Table 21. The steps before this step are the same as those before the fuzzy TOPSIS method to calculate the distance. Secondly, in order to obtain the concordance matrix, we use Equation (19) to calculate the preference value between two given alternatives on each criterion, and the results are shown in Table 22. Thirdly, we use Equation (20) to get the discordance matrix as shown in Table 23. Fourthly, we obtain the Boolean matrices B and H based on the minimum concordance level and minimum discordance level by using Equations (21) and (22), respectively. The Boolean matrices B and H are shown in Table 24 and Table 25 respectively. Fifthly, we use Equation (23) to obtain the global matrix Z as shown in Table 26. Finally, we get the results of the fuzzy ELECTRE I, as shown in Table 27. 

It can be concluded from the global matrix Z that A2, A5 and A9 are the three alternatives with the most advantages over other alternatives. Moreover, we can conclude from the outranking relations among all the alternatives that A2, A5 and A9 are incomparable. Therefore, A2, A5 and A9 constitute a set of best alternatives.

### 4.5. Discussion

In this section, we compare the results of the application of the fuzzy TOPSIS method and the ELECTRE I method to the green supplier selection of a Chinese Internet company, as shown in Table 28.

We also use a line chart to compare the ranking of the green suppliers of the internet company under the two methods, as shown in Figure 5. The figure more vividly shows the similarity of the evaluation of ten suppliers under the two methods.

By considering the rankings of each alternative of the two methods in Table 28, it is found that the rankings of the ten alternatives are almost the same; in this case, A9 is the first and A10 is the tenth in the decision-making problem of green supplier selection. Generally, enterprises can choose more than one supplier in order to reduce the risk brought by suppliers. Therefore, A2 can be included in the supplier range selected by the company as the second of all alternatives. Comparing the TOPSIS method and the ELECTRE I method, there are the following similarities and differences: 

(1) Both are based on different criteria and the relative importance of the criteria in the linguistic variables, and the scheme is intuitively fuzzy scored, then, the true value membership and non-truth value membership function. 

(2) Both are based on the concept of distance measurement in an intuitionistic fuzzy environment. 

(3) The TOPSIS method provides full compensation in the group decision-making process; however, the ELECTRE method provides partial compensation from the perspective of criterion information processing [58].

(4) The TOPSIS method is a derivation of a preference rank order based on approaching ideal solution technology; but, the ELECTRE method is a derivation of recommendations based on outranking relations.

(5) The goal of the TOPSIS method is ranking problematic; the goal of the ELECTRE method is choice problematic [57]. The TOPSIS method can provide a selection sequence of alternatives; the ELECTRE method will provide a set of good alternatives.

In our research process, at least three advantages of the TOPSIS method are fully reflected and applied [41,42]: 

(1) A reasonable logical fit with the basic principles of human selection;

(2) A scalar value that accounts for both the best and worst alternatives simultaneously; 

(3) A simple calculation process that is easy to program into a spreadsheet.

Therefore, the results of our study are highly credible.

## 5. Conclusions 

From the review of the current literature, it can be seen that GSCM has become a topic that enterprises must pay attention to. GSCM practice will help companies improve their business performance while responding to the background of green development. An important part of the company’s GSCM practice—how to evaluate suppliers and select the best green suppliers is critical.

This paper studies the selection of green suppliers in the language environment of triangular fuzzy numbers based on GSCM practice. The framework of a green supplier selection method for enterprises is presented. In order to make a reasonable and reliable evaluation of alternative suppliers, this paper selects 12 criteria that affect supplier GSCM practices based on the literature review and expert’s opinions. Using the fuzzy TOPSIS method and the ELECTRE method, the results of the two methods are compared to obtain the best green suppliers. Applying the framework to the green supplier selection of a Chinese internet company, the results show that the main influential criteria for CSCM practices are management support for green supply chain management, use of environmentally friendly materials, following legal environmental requirements and policies, reducing the use of harmful substances, developing ecological products and sustainable recycling design. The fundamental requirement for achieving really green development is the support of management for GSCM. In addition, our new criterion 12 (quality after-sales service) also affects the selection of green suppliers. In China, companies have gradually realized the importance of supply chain management and the importance of after-sales service. Yet, few companies combine supplier selection with their after-sales services for governance. After-sales service is not only a service strategy formulated by manufacturers to distributors and retail enterprises, or a service strategy of sellers to consumers, but also a service of suppliers to manufacturers. The complexity of supply chain management is obvious. By reducing problems at the source, the entire supply chain can run more smoothly. The criteria are divided into benefit and cost, and 10 suppliers are evaluated by 12 criteria. The alternative which is close to the positive-ideal solution but far away from the negative-ideal solution was selected as the best alternative. Finally, we analyzed and found that alternative 9 is the best selection and that alternative 2 can also be included in the company’s selection of green suppliers

The superposition of the two methods improves the reliability of the results. However, this paper does have some limitations. This paper proposes an evaluation framework for selecting green suppliers and applies it to practical cases, but it is not widely used for the time being. In addition, this paper also has shortcomings in methods. For the two methods of fuzzy TOPSIS and ELECTRE, the application of this paper will result in the difficulty of data processing in the case of many alternative suppliers and many evaluation criteria. We will make innovations in the application of the two methods in future research, making them more convenient, intuitive and effective, and the results more precise, so as to improve the application of the framework. Our future research methods may include the use of Intuitionistic fuzzy sets and Pythagorean sets, and then, compare the results obtained by these methods with the results obtained in this work. The improvement of the solution method of the supplier selection problem and the development of a more efficient and high-quality group decision support system under the fuzzy environment are the directions of future research topics.

## Figures and Tables

**Figure 1 ijerph-17-03268-f001:**
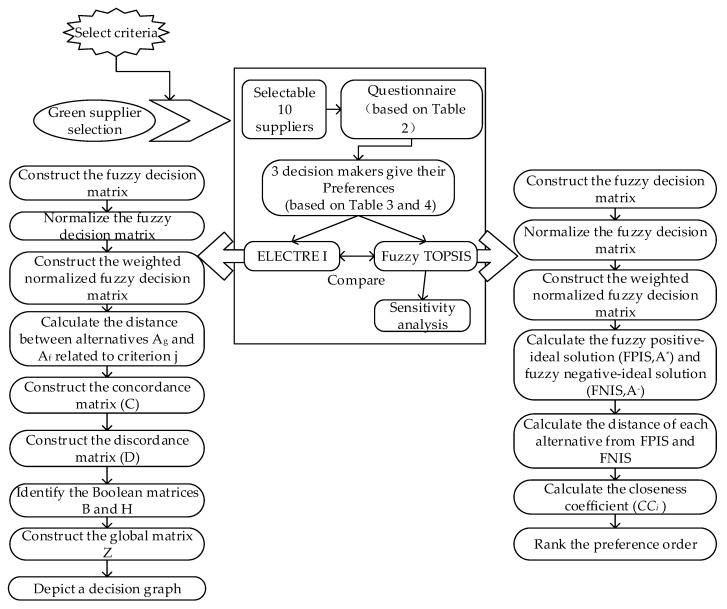
Proposed framework used in this paper.

**Figure 2 ijerph-17-03268-f002:**
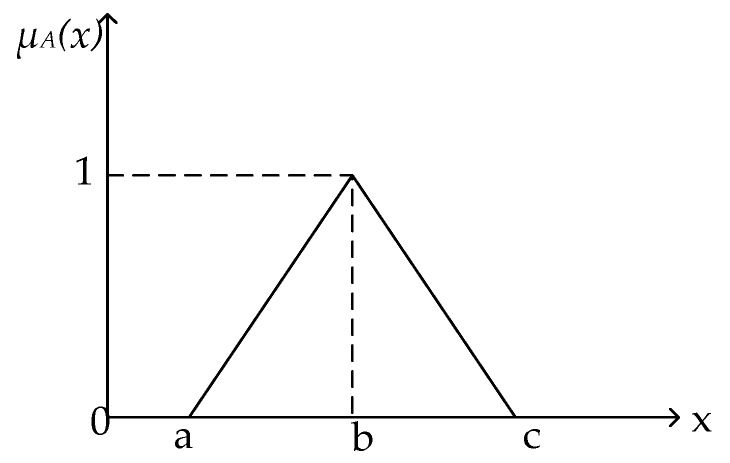
Membership function of triangular fuzzy number A.

**Figure 3 ijerph-17-03268-f003:**
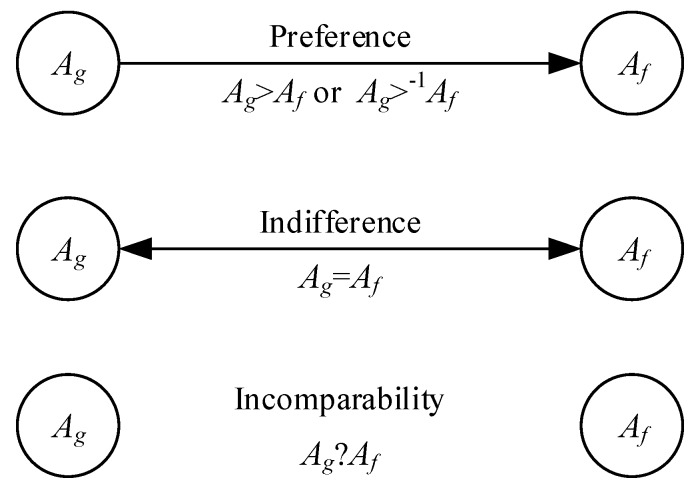
The graphical representation of the binary relations (>, =, ?) used in the decision graph.

**Figure 4 ijerph-17-03268-f004:**
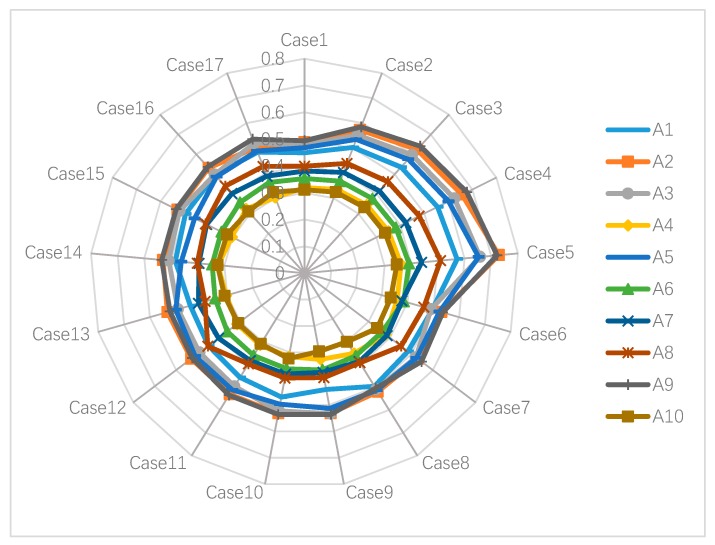
Results of sensitivity analysis.

**Figure 5 ijerph-17-03268-f005:**
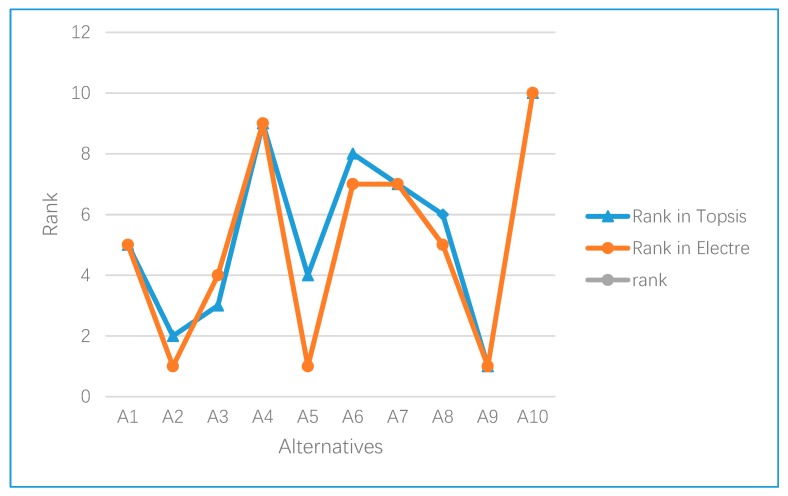
The pictorial representation of the TOPSIS and ELECTRE I rankings.

**Table 1 ijerph-17-03268-t001:** Top 10 cited works in green supply chain management.

Author	Title	Purpose
Genovese (2017)	Sustainable supply chain management and the transition towards a circular economy: Evidence and some applications.	By comparing a series of performance indicators of traditional and circular production systems, this paper proposes that integrating circular economy principles in sustainable supply chain management can provide significant environmental advantages.
Qin, J. D. (2017)	An extended TODIM multi-criteria group decision-making method for green supplier selection in interval type-2 fuzzy environment.	This paper proposes an extended TODIM method based on the prospect theory to solve the MCGDM problem in the IT2FSs environment, and gives its application to the problem of green supplier selection.
Li, B. (2016)	Pricing policies of a competitive dual-channel green supply chain.	This paper introduces e-commerce into green supply chain management, and discusses the pricing and greening strategies of chain members in centralized and decentralized cases using the Stackelberg game model.
Liou, J. J. H. (2016)	New hybrid COPRAS-G MADM Model for improving and selecting suppliers in green supply chain management.	This study proposes a new hybrid model that uses Decision Experiment and Evaluation Lab (DEMATEL) technology to build the relationship between the criteria, thereby constructing an influential network relationship diagram (INRM) that addresses various standards and Decision makers have fuzzy dependencies between information.
Dubey, R. (2017)	Sustainable supply chain management: framework and further research directions.	This article presents the idea of applying Total Interpretive Structural Modeling (TISM) in sustainable supply chain management (SSCM). This paper advocates an alternative approach to solving problems related to the SSCM driver.
Rajeev, A. (2017)	Evolution of sustainability in supply chain management: A literature review.	This article attempts to propose a conceptual framework by analyzing trends between industries, economies (targeting heavily polluting industries, especially those in emerging economies), and using various methods to understand the evolution of sustainability issues, and classify the various factors on the triple bottom line pillar of sustainability issues in the supply chain.
Chiappetta Jabbour, C. J. (2016)	Green Human Resource Management and Green Supply Chain Management: linking two emerging agendas.	This paper proposed a synergistic and integrative framework for the GHRM-GSCM relationship and integrative and proposed a research agenda for it. On this basis, this study highlights the impact of GHRM-GSCM integration on academics, managers, and practitioners in the areas of organizational sustainability and truly sustainable supply chains.
Formentini, M. (2016)	Corporate sustainability approaches and governance mechanisms in sustainable supply chain management.	This paper proposed an empirical investigation by analyzing seven case studies through the lenses of contingency theory, the strategic alignment perspective and the resource-based view of organizations. The report explains how the corporate sustainability approach is implemented and is consistent with governance mechanisms at the supply chain level.
Quarshie, A. M. (2016)	Sustainability and corporate social responsibility in supply chains: The state of research in supply chain management and business ethics journals.	This paper examines and compares existing research and knowledge creation on corporate social responsibility (CSR) issues in sustainable development and supply chains, and proposes a future research agenda that will link disciplines and highlight key areas that will benefit from further research.
Geng, R. (2017)	The relationship between green supply chain management and performance: A meta-analysis of empirical evidences in Asian emerging economies.	The study found that GSCM practices led to better performance in terms of economics, environment, operations, and social performance. The paper adopted meta-analysis for verifying GSCM practice-performance relationships in the manufacturing sector of AEE.

**Table 2 ijerph-17-03268-t002:** Selected criteria in this paper.

Code	Criteria	Definition	Category
C1	Management support for supply chain management	Management is very important to adopt green supply chain management.	B
C2	Follow legal environmental requirements and policies	Comply with legal requirements and environmental management standards.	B
C3	Passed ISO14001 certification	The ISO 14000 series of standards is an environmental management system standard developed by the International Organization for Standardization.	B
C4	Use environmentally friendly technologies and equipment	Make it energy efficient and produce clean products.	B
C5	Developing ecological products	Develop eco-products to reduce environmental impact.	B
C6	Use environmentally friendly materials	Produce products or packaged products using environmentally friendly materials or Recyclable material.	B
C7	Reduce the use of harmful substances	Control the range of use of hazardous materials when producing products.	B
C8	Lean management	Create maximum value with minimal resources	B
C9	Sustainable recycling design	A systematic design based on the comprehensive standards of economy, ecology and society	B
C10	Reasonable inventory management	Through reasonable inventory management, companies can sell excess materials or products to restore their investment.	B
C11	Conduct internal environmental management evaluation of suppliers	Provides a way to check the environmental performance of suppliers and ensure that they comply with environmental management standards.	B
C12	Quality after-sales service	Add additional services to build a good customer sense and achieve sustainable business development.	B

**Table 3 ijerph-17-03268-t003:** Linguistic variables and fuzzy ratings of the alternative.

Linguistic Terms	Fuzzy Numbers
Very poor (VP)	(0,0,1)
Poor (P)	(0,1,3)
Medium poor (MP)	(1,3,5)
Fair (F)	(3,5,7)
Medium good (MG)	(5,7,9)
Good (G)	(7,9,10)
Very good (VG)	(9,10,10)

**Table 4 ijerph-17-03268-t004:** Linguistic variables and fuzzy ratings of the criteria.

Linguistic Terms	Fuzzy Numbers
Very low (VL)	(0,0.2,0.4)
Low (L)	(0.2,0.4,0.6)
Medium (M)	(0.4,0.6,0.8)
High (H)	(0.6,0.8,1)
Very high (VH)	(0.8,0.9,1)

**Table 5 ijerph-17-03268-t005:** Decision makers’ assessment of 12 standard linguistic variables.

DMs	C1	C2	C3	C4	C5	C6	C7	C8	C9	C10	C11	C12
DM1	VH	H	M	VH	H	VH	H	L	H	M	H	M
DM2	VH	H	VH	M	H	H	VH	M	H	M	H	H
DM3	VH	VH	M	H	H	VH	H	VH	H	L	H	VH

**Table 6 ijerph-17-03268-t006:** Decision maker 1 (DM1) evaluates the alternatives.

Criteria	A1	A2	A3	A4	A5	A6	A7	A8	A9	A10
C1	VG	G	VG	F	VG	G	MP	VG	G	P
C2	G	G	G	G	VG	MP	P	G	VG	F
C3	VG	MG	MG	MP	VG	MP	MP	MP	VG	P
C4	F	G	VG	MP	VG	VP	MP	P	VG	VP
C5	G	G	G	P	F	VP	VP	P	VG	MP
C6	MP	G	F	MP	VG	MP	P	P	VG	MP
C7	F	G	MG	VP	VG	MP	F	G	VG	VP
C8	P	G	MG	VP	G	P	F	VP	G	P
C9	G	G	G	VP	F	P	F	F	VG	F
C10	VG	G	VG	P	P	P	MP	F	VG	P
C11	G	G	MG	F	G	F	F	G	VG	P
C12	G	F	G	P	G	F	P	F	VG	MP

**Table 7 ijerph-17-03268-t007:** Decision maker 2 (DM2) evaluates the alternatives.

Criteria	A1	A2	A3	A4	A5	A6	A7	A8	A9	A10
C1	G	VG	VG	G	VG	F	P	VG	VG	F
C2	G	VG	G	G	VG	F	MP	VG	VG	F
C3	VG	G	G	G	VG	F	MP	MP	VG	P
C4	P	G	VG	P	VG	F	P	P	VG	VP
C5	F	VG	VG	MP	F	P	P	F	VG	MP
C6	F	VG	MG	P	G	MP	P	F	G	VP
C7	F	G	F	P	G	MP	F	VG	G	MP
C8	F	VG	F	P	MG	MP	F	P	MG	MP
C9	G	G	MG	P	P	P	P	MP	G	F
C10	G	G	G	VP	P	MP	F	F	G	F
C11	G	G	F	P	G	P	F	MG	VG	MP
C12	G	MG	G	VP	MG	F	F	MG	G	MP

**Table 8 ijerph-17-03268-t008:** Decision maker 3 (DM3) evaluates the alternatives.

Criteria	A1	A2	A3	A4	A5	A6	A7	A8	A9	A10
C1	VG	VG	VG	G	VG	F	F	VG	VG	MG
C2	MG	G	VG	G	VG	F	F	G	VG	F
C3	G	VG	G	G	G	F	MP	MP	VG	VP
C4	P	VG	VG	F	G	MG	MP	F	G	VP
C5	F	G	VG	MP	VG	MG	F	MP	G	MP
C6	MG	G	G	P	G	F	F	MP	G	P
C7	MP	G	F	P	G	F	F	VG	MG	P
C8	MP	VG	F	P	MG	MP	MG	MP	MG	VP
C9	MG	G	G	F	F	MP	MG	MP	G	P
C10	G	G	G	VP	F	P	MG	F	G	MP
C11	G	VG	MG	VP	F	MP	F	G	G	P
C12	MG	MG	G	VP	F	MP	F	G	VG	MP

**Table 9 ijerph-17-03268-t009:** Fuzzy aggregated weights of each criterion.

Criteria	Fuzzy Weights
C1	(0.8,0.9,1)
C2	(0.6,0.833,1)
C3	(0.4,0.7,1)
C4	(0.4,0.767,1)
C5	(0.6,0.8,1)
C6	(0.6,0.867,1)
C7	(0.6,0.833,1)
C8	(0.4,0.7,1)
C9	(0.6,0.8,1)
C10	(0.2,0.533,0.8)
C11	(0.6,0.8,1)
C12	(0.4,0.7,1)

**Table 10 ijerph-17-03268-t010:** Fuzzy aggregated decision matrix.

**Criteria**	**A1**	**A2**	**A3**	**A4**	**A5**
C1	(7,9.667,10)	(7,9.667,10)	(9,10,10)	(3,7.667,10)	(9,10,10)
C2	(5,8.333,10)	(7,9.333,10)	(7,9.333,10)	(7,9,10)	(9,10,10)
C3	(7,9.667,10)	(5,8.667,10)	(5,8.333,10)	(1,7.333,10)	(7,9.667,10)
C4	(0,2.333,7)	(7,9.333,10)	(9,10,10)	(0,3,7)	(7,9.667,10)
C5	(3,6.333,10)	(7,9.667,10)	(7,9.667,10)	(0,2.333,5)	(3,10,10)
C6	(1,5,9)	(7,9.333,10)	(3,7,10)	(0,1.667,5)	(7,9.333,10)
C7	(1,4.333,7)	(7,9,10)	(3,5.667,9)	(0,1.667,5)	(7,9.333,10)
C8	(0,3,7)	(7,9.667,10)	(3,5.667,9)	(0,0.667,3)	(5,7.667,10)
C9	(5,8.333,10)	(7,9,10)	(5,8.333,10)	(0,2,7)	(0,3.667,7)
C10	(7,9.333,10)	(7,9,10)	(7,9.333,10)	(0.0.333,3)	(0,2.333,7)
C11	(7,9,10)	(7,9.333,10)	(3,7.667,9)	(0,2,7)	(3,5.667,10)
C12	(5,8.333,10)	(3,6.333,9)	(7,9,10)	(0.0.333,3)	(3,7,10)
**Criteria**	**A6**	**A7**	**A8**	**A9**	**A10**
C1	(3,6.333,10)	(0,3,5)	(9,10,10)	(7,9.667,10)	(0,4.333,9)
C2	(1,4.333,7)	(0,3,7)	(7,9.667,10)	(9,10,10)	(3,5,7)
C3	(1,4.333,7)	(1,3,5)	(1,3,5)	(9,10,10)	(0,1.667,3)
C4	(0,4,9)	(0,2.333,5)	(0,2.333,7)	(7,9.667,10)	(0,0,1)
C5	(0,2.667,9)	(0,2,7)	(0,3,7)	(7,9.667,10)	(1,3,5)
C6	(1,3.667,7)	(0,2.333,7)	(0,3,7)	(7,9.333,10)	(0,1.333,5)
C7	(1,3.667,7)	(3,5,7)	(7,9.667,10)	(5,8.667,10)	(0,1.333,5)
C8	(0,2.333,5)	(3,5.667,9)	(0,1.333,5)	(5,7.667,10)	(0,1.333,5)
C9	(0,1.667,5)	(0,4.333,9)	(1,3.667,7)	(7,9.333,10)	(0,3.667,7)
C10	(0,1.667,5)	(1,5,9)	(3,5,7)	(7,9.333,10)	(0,3,7)
C11	(0,3,7)	(3,5,7)	(5,8.333,10)	(7,9.667,10)	(0,1.667,5)
C12	(1,4.333,7)	(0.3.667,7)	(3,7,10)	(7,9.667,10)	(1,3,5)

**Table 11 ijerph-17-03268-t011:** The category of each criterion.

Criteria	Weights	Category	Criteria	Weights	Category
C1	(0.8,0.9,1)	B	C7	(0.6,0.833,1)	B
C2	(0.6,0.833,1)	B	C8	(0.4,0.7,1)	B
C3	(0.4,0.7,1)	B	C9	(0.6,0.8,1)	B
C4	(0.4,0.767,1)	B	C10	(0.2,0.533,0.8)	B
C5	(0.6,0.8,1)	B	C11	(0.6,0.8,1)	B
C6	(0.6,0.867,1)	B	C12	(0.4,0.7,1)	B

**Table 12 ijerph-17-03268-t012:** Normalized fuzzy aggregated decision matrix.

**Criteria**	**A1**	**A2**	**A3**	**A4**	**A5**
C1	(0.7,0.967,1)	(0.7,0.967,1)	(0.9,1,1)	(0.3,0.767,1)	(0.9,1,1)
C2	(0.5,0.833,1)	(0.7,0.933,1)	(0.7,0.933,1)	(0.7,0.9,1)	(0.9,1,1)
C3	(0.7,0.967,1)	(0.5,0.867,1)	(0.5,0.833,1)	(0.1,0.733,1)	(0.7,0.967,1)
C4	(0,0.233,0.7)	(0.7,0.933,1)	(0.9,1,1)	(0,0.3,0.7)	(0.7,0.967,1)
C5	(0.3,0.633,1)	(0.7,0.967,1)	(0.7,0.967,1)	(0,0.233,0.5)	(0.3,1,1)
C6	(0.1,0.5,0.9)	(0.7,0.933,1)	(0.3,0.7,1)	(0,0.167,0.5)	(0.7,0.933,1)
C7	(0.1,0.433,0.7)	(0.7,0.9,1)	(0.3,0.567,0.9)	(0,0.167,0.5)	(0.7,0.933,1)
C8	(0,0.3,0.7)	(0.7,0.967,1)	(0.3,0.567,0.9)	(0,0.067,0.3)	(0.5,0.767,1)
C9	(0.5,0.833,1)	(0.7,0.9,1)	(0.5,0.833,1)	(0,0.2,0.7)	(0,0.367,0.7)
C10	(0.7,0.933,1)	(0.7,0.9,1)	(0.7,0.933,1)	(0.0.033,0.3)	(0,0.233,0.7)
C11	(0.7,0.9,1)	(0.7,0.933,1)	(0.3,0.767,0.9)	(0,0.2,0.7)	(0.3,0.767,1)
C12	(0.5,0.833,1)	(0.3,0.633,0.9)	(0.7,0.9,1)	(0.0.033,0.3)	(0.3,0.7,1)
**Criteria**	**A6**	**A7**	**A8**	**A9**	**A10**
C1	(0.3,0.633,1)	(0,0.333,0.5556)	(0.9,1,1)	(0.7,0.967,1)	(0,0.481,1)
C2	(0.1,0.433,0.7)	(0,0.333,0.7778)	(0.7,0.967,1)	(0.9,1,1)	(0.333,0.556,0.778)
C3	(0.1,0.433,0.7)	(0.333,0.333,0.556)	(0.1,0.3,0.5)	(0.9,1,1)	(0,0.185,0.3333)
C4	(0,0.4,0.9)	(0,0.259,0.556)	(0,0.233,0.7)	(0.7,0.967,1)	(0,0,0.333)
C5	(0,0.267,0.9)	(0,0.222,0.778)	(0,0.3,0.7)	(0.7,0.967,1)	(0.333,0.333,0.556)
C6	(0.1,0.367,0.7)	(0,0.259,0.778)	(0,0.3,0.7)	(0.7,0.933,1)	(0,0.148,0.556)
C7	(0.1,0.367,0.7)	(0.333,0.556,0.778)	(0.7,0.967,1)	(0.5,0.867,1)	(0,0.148,0.5556)
C8	(0,0.233,0.5)	(0.333,0.630,1)	(0,0.133,0.5)	(0.5,0.767,1)	(0,0.148,0.556)
C9	(0,0.167,0.5)	(0,0.481,1)	(0.1,0.367,0.7)	(0.7,0.933,1)	(0,0.407,0.778)
C10	(0,0.167,0.5)	(0.333,0.556,1)	(0.3,0.5,0.7)	(0.7,0.933,1)	(0,0.333,0.778)
C11	(0,0.3,0.7)	(0.333,0.556,0.778)	(0.5,0.833,1)	(0.7,0.967,1)	(0,0.185,0.556)
C12	(0.1,0.433,0.7)	(0.0.407,0.778)	(0.3,0.7,1)	(0.7,0.967,1)	(0.333,0.333,0.556)

**Table 13 ijerph-17-03268-t013:** Weighted normalized alternatives.

	A1	A2	A3	A4	A5	A6	A7	A8	A9	A10	FPIS(A*)	FNIS(A-)
C1	(0.56,0.870,1)	(0.56,0.870,1)	(0.72,0.9,1)	(0.24,0.690,1,)	(0.72,0.9,1)	(0.24,0.570,1)	(0,0.300,0.556)	(0.72,0.9,1)	(0.56,0.870,1)	(0,0.433,1)	1	0
C2	(0.3,0.694,1)	(0.42,0.777,1)	(0.42,0.777,1)	(0.42,0.750,1)	(0.54,0.833,1)	(0.06,0.360,0.7)	(0,0.277, 0.778)	(0.42,0.806,1)	(0.54,0.833,1)	(0.200,0.46,0.778)	1	0
C3	(0.28,0.677,1)	(0.2,0.607,1)	(0.2,0.583,1)	(0.04,0.513,1)	(0.28,0.677,1)	(0.04,0.303,0.7)	(0.133,0.233,0.556)	(0.04,0.21,0.5)	(0.36,0.7,1)	(0,0.130,0.333)	1	0
C4	(0,0.179,0.7)	(0.28,0.716,1)	(0.36,0.767,1)	(0,0.230,0.7)	(0.28,0.742,1)	(0,0.307,0.9)	(0,0.199,0.556)	(0,0.179,0.7)	(0.28,0.742,1)	(0,0,0.333)	1	0
C5	(0.18,0.506,1)	(0.42,0.774,1)	(0.42,0.774,1)	(0,0.186,0.5)	(0.18,0.8,1)	(0,0.214,0.9)	(0,0.178, 0.779)	(0,0.24,0.7)	(0.42,0.774,1)	(0.200,0.266,0.556)	1	0
C6	(0.06,0.434,0.9)	(0.42,0.809,1)	(0.18,0.607,1)	(0,0.145,0.5)	(0.42,0.809,1)	(0.06,0.318,0.7)	(0,0.225,0.778)	(0,0.260,0.7)	(0.42,0.809,1)	(0,0.128,0.556)	1	0
C7	(0.06,0.361,0.7)	(0.42,0.750,1)	(0.18,0.472,0.9)	(0,0.139,0.5)	(0.42,0.777,1)	(0.06,0.306,0.7)	(0.200,0.463,0.778)	(0.42,0.806,1)	(0.3,0.722,1)	(0,0.123,0.556)	1	0
C8	(0,0.21,0.7)	(0.28,0.677,1)	(0.12,0.397,0.9)	(0,0.047,0.3)	(0.2,0.537,1)	(0,0.163,0.5)	(0.133,0.441,1)	(0,0.093,0.5)	(0.2,0.537,1)	(0,0.104,0.556)	1	0
C9	(0.3,0.666,1)	(0.42,0.72,1)	(0.3,0.666,1)	(0,0.16,0.7)	(0,0.294,0.7)	(0,0.134,0.5)	(0,0.385,1)	(0.06,0.294,0.7)	(0.42,0.746,1)	(0,0.326,0.778)	1	0
C10	(0.14,0.497,0.8)	(0.14,0.480,0.8)	(0.14,0.497,0.8)	(0,0.018,0.24)	(0,0.124, 0.56)	(0,0.089,0.4)	(0.067,0.296,0.8)	(0.06,0.267,0.56)	(0.14,0.497,0.8)	(0,0.178,0.622)	1	0
C11	(0.42,0.72,1)	(0.42,0.746,1)	(0.18,0.614,0.9)	(0,0.16,0.7)	(0.18,0.614,0.9)	(0,0.24,0.7)	(0.200,0.445,0.778)	(0.3,0.666,1)	(0.42, 0.774,1)	(0,0.148,0.556)	1	0
C12	(0.2,0.583,1)	(0.12,0.443,0.9)	(0.28,0.63,1)	(0,0.023,0.3)	(0.12,0.49,1)	(0.04,0.303,0.7)	(0,0.285,0.778)	(0.12,0.49,1)	(0.28,0.677,1)	(0.133,0.233,0.556)	1	0

FPIS(A*): fuzzy positive-ideal solution; FNIS(A-): fuzzy negative-ideal solution.

**Table 14 ijerph-17-03268-t014:** Distance *d_v_* (*A_i_,*
*A**) for alternatives.

Criteria	A1	A2	A3	A4	A5	A6	A7	A8	A9	A10
C1	0.2649	0.2649	0.1717	0.4739	0.1717	0.5041	0.7499	0.1717	0.2649	0.6637
C2	0.4411	0.3588	0.3588	0.3646	0.2825	0.679	0.7239	0.3531	0.2825	0.5718
C3	0.4556	0.5146	0.5209	0.6215	0.4556	0.7065	0.7158	0.7737	0.4081	0.8567
C4	0.7668	0.4469	0.3932	0.749	0.4416	0.7048	0.7829	0.7668	0.4416	0.9028
C5	0.5527	0.3594	0.3594	0.7985	0.4873	0.7366	0.7582	0.7456	0.3594	0.6772
C6	0.6361	0.3526	0.525	0.8126	0.3526	0.6925	0.7416	0.7388	0.3526	0.8078
C7	0.6787	0.3646	0.566	0.8147	0.3588	0.6965	0.5709	0.3531	0.4349	0.8096
C8	0.7559	0.4556	0.6186	0.8941	0.5337	0.8063	0.5956	0.8312	0.5337	0.8165
C9	0.4478	0.3718	0.4478	0.7736	0.7277	0.8165	0.6778	0.7005	0.3656	0.7079
C10	0.5867	0.5916	0.5867	0.9205	0.8085	0.8544	0.6846	0.7336	0.5867	0.7786
C11	0.3718	0.3656	0.5264	0.7736	0.5264	0.7456	0.5766	0.4478	0.3594	0.8006
C12	0.5209	0.6041	0.4674	0.9027	0.5872	0.7065	0.7212	0.5872	0.4556	0.7158
Σ	6.479	5.0505	5.5419	8.8993	5.7336	8.6493	8.299	7.2031	4.845	9.109

**Table 15 ijerph-17-03268-t015:** Distance *d_v_* (*A_i_*, *A^−^*) for alternatives.

Criteria	A1	A2	A3	A4	A5	A6	A7	A8	A9	A10
C1	0.8308	0.8308	0.8810	0.7150	0.8810	0.6788	0.3648	0.8810	0.8308	0.6291
C2	0.7238	0.7703	0.7703	0.7613	0.8135	0.4558	0.4768	0.7802	0.8135	0.5344
C3	0.7157	0.6852	0.6782	0.6493	0.7157	0.4410	0.3564	0.3140	0.7348	0.2064
C4	0.4171	0.7283	0.7567	0.4254	0.7369	0.5490	0.3409	0.4171	0.7369	0.1923
C5	0.6553	0.7693	0.7693	0.3080	0.7466	0.5341	0.4613	0.4272	0.7693	0.3741
C6	0.5779	0.7812	0.6833	0.3006	0.7812	0.4452	0.4676	0.4311	0.7812	0.3294
C7	0.4560	0.7613	0.5959	0.2996	0.7703	0.4424	0.5353	0.7802	0.7329	0.3288
C8	0.4219	0.7157	0.5721	0.1753	0.6654	0.3036	0.6357	0.2936	0.6654	0.3266
C9	0.7150	0.7516	0.7150	0.4146	0.4383	0.2989	0.6187	0.4397	0.7600	0.4870
C10	0.5497	0.5447	0.5497	0.1390	0.3311	0.2366	0.4940	0.3599	0.5497	0.3735
C11	0.7516	0.7600	0.6375	0.4146	0.6375	0.4272	0.5302	0.7150	0.7693	0.3322
C12	0.6782	0.5833	0.7013	0.1737	0.6467	0.4410	0.4784	0.6467	0.7157	0.3564
Σ	7.493	8.6817	8.3103	4.7764	8.1642	5.2536	5.7601	6.4857	8.8595	4.4702

**Table 16 ijerph-17-03268-t016:** Closeness coefficient (CCi).

Distance and Closeness Coefficent	A1	A2	A3	A4	A5	A6	A7	A8	A9	A10
d*	6.479	5.051	5.542	8.899	5.734	8.649	8.299	7.203	4.845	9.109
d^−^	7.493	8.682	8.310	4.776	8.164	5.254	5.760	6.486	8.860	4.470
CCi	0.536	0.632	0.600	0.349	0.587	0.378	0.410	0.474	0.646	0.329

**Table 17 ijerph-17-03268-t017:** Ranking of alternatives.

Alternatives	Ranking
A1	5
A2	2
A3	3
A4	9
A5	4
A6	8
A7	7
A8	6
A9	1
A10	10

**Table 18 ijerph-17-03268-t018:** Ranking of criteria’s weights.

Criteria	Weights	Ranking
C1	(0.8,0.9,1)	1
C2	(0.6,0.833,1)	3
C3	(0.4,0.7,1)	6
C4	(0.4,0.767,1)	5
C5	(0.6,0.8,1)	4
C6	(0.6,0.867,1)	2
C7	(0.6,0.833,1)	3
C8	(0.4,0.7,1)	6
C9	(0.6,0.8,1)	4
C10	(0.2,0.533,0.8)	7
C11	(0.6,0.8,1)	4
C12	(0.4,0.7,1)	6

**Table 19 ijerph-17-03268-t019:** Details for sensitivity analysis.

Case	Description (change made in W_jt_), j = C1,C2,...,C12, t = 1,2,3
Case1	W_c1–c12_ = (0,0.2,0.4)
Case2	W_c1–c12_ = (0.2,0.4,0.6)
Case3	W_c1–c12_ = (0.4,0.6,0.8)
Case4	W_c1–c12_ = (0.6,0.8,1)
Case5	W_c1–c12_ = (0.8,0.9,1)
Case6	W_c1_ = (0.8,0.9,1), W_c2–c12_ = (0,0.2,0.4)
Case7	W_c2_ = (0.8,0.9,1), W_c1,c3–c12_ = (0,0.2,0.4)
Case8	W_c3_ = (0.8,0.9,1), W_c1–c2,c4–c12_ = (0,0.2,0.4)
Case9	W_c4_ = (0.8,0.9,1), W_c1–c3,c5–c12_ = (0,0.2,0.4)
Case10	W_c5_ = (0.8,0.9,1), W_c1–c4,c6–c12_ = (0,0.2,0.4)
Case11	W_c6_ = (0.8,0.9,1), W_c1–c5,c7–c12_ = (0,0.2,0.4)
Case12	W_c7_ = (0.8,0.9,1), W_c1–c6,c8–c12_ = (0,0.2,0.4)
Case13	W_c8_ = (0.8,0.9,1), W_c1–c7,c9–c12_ = (0,0.2,0.4)
Case14	W_c9_ = (0.8,0.9,1), W_c1–c8,c10–c12_ = (0,0.2,0.4)
Case15	W_c10_ = (0.8,0.9,1), W_c1–c9,c11–c12_ = (0,0.2,0.4)
Case16	W_c11_ = (0.8,0.9,1), W_c1–c10,c12_ = (0,0.2,0.4)
Case17	W_c12_ = (0.8,0.9,1), W_c1–c11_ = (0,0.2,0.4)

**Table 20 ijerph-17-03268-t020:** Results of sensitivity analysis.

Case	A1	A2	A3	A4	A5	A6	A7	A8	A9	A10	Ranking
Case 1	0.449	0.488	0.478	0.319	0.468	0.353	0.381	0.399	0.494	0.311	A9 > A2 > A3 > A5 > A1 > A8 > A7 > A6 > A4 > A10
Case 2	0.503	0.573	0.552	0.335	0.535	0.368	0.403	0.439	0.585	0.324	A9 > A2 > A3 > A5 > A1 > A8 > A7 > A6 > A4 > A10
Case 3	0.535	0.624	0.596	0.344	0.575	0.376	0.415	0.462	0.641	0.332	A9 > A2 > A3 > A5 > A1 > A8 > A7 > A6 > A4 > A10
Case 4	0.555	0.657	0.625	0.350	0.601	0.381	0.423	0.478	0.677	0.336	A9 > A2 > A3 > A5 > A1 > A8 > A7 > A6 > A4 > A10
Case 5	0.572	0.729	0.658	0.362	0.653	0.392	0.439	0.510	0.722	0.346	A2 > A9 > A3 > A5 > A1 > A8 > A7 > A6 > A4 > A10
Case 6	0.496	0.531	0.491	0.360	0.524	0.385	0.378	0.463	0.536	0.335	A9 > A2 > A5 > A1 > A3 > A8 > A6 > A7 > A4 > A10
Case 7	0.484	0.531	0.521	0.380	0.524	0.363	0.386	0.452	0.547	0.340	A9 > A2 > A5 > A3 > A1 > A8 > A7 > A6 > A4 > A10
Case 8	0.496	0.520	0.509	0.352	0.513	0.363	0.388	0.393	0.505	0.301	A2 > A9 > A3 > A5 > A1 > A8 > A7 > A6 > A4 > A10
Case 9	0.441	0.532	0.533	0.327	0.513	0.367	0.376	0.396	0.536	0.297	A9 > A3 > A2 > A5 > A1 > A8 > A7 > A6 > A4 > A10
Case 10	0.470	0.532	0.522	0.317	0.497	0.363	0.383	0.398	0.536	0.324	A9 > A2 > A3 > A5 > A1 > A8 > A7 > A6 > A10 > A4
Case 11	0.457	0.531	0.497	0.315	0.512	0.361	0.384	0.398	0.535	0.310	A9 > A2 > A5 > A3 > A1 > A8 > A7 > A6 > A4 > A10
Case 12	0.449	0.530	0.491	0.315	0.512	0.361	0.402	0.452	0.525	0.310	A2 > A9 > A5 > A3 > A8 > A1 > A7 > A6 > A4 > A10
Case 13	0.443	0.532	0.491	0.304	0.499	0.347	0.411	0.386	0.522	0.310	A2 > A9 > A5 > A3 > A1 > A7 > A8 > A6 > A10 > A4
Case 14	0.484	0.530	0.509	0.324	0.462	0.346	0.396	0.402	0.535	0.326	A9 > A2 > A3 > A1 > A5 > A8 > A7 > A6 > A10 > A4
Case 15	0.496	0.530	0.521	0.303	0.458	0.346	0.408	0.413	0.535	0.323	A9 > A2 > A3 > A1 > A5 > A8 > A7 > A6 > A10 > A4
Case 16	0.495	0.531	0.497	0.324	0.487	0.357	0.402	0.439	0.536	0.311	A9 > A2 > A3 > A1 > A5 > A8 > A7 > A6 > A4 > A10
Case 17	0.484	0.503	0.520	0.303	0.489	0.363	0.388	0.427	0.536	0.324	A9 > A3 > A2 > A5 > A1 > A8 > A7 > A6 > A10 > A4

**Table 21 ijerph-17-03268-t021:** The distance between two alternatives g and f with respect to each criterion.

	**X_11_**	**X_21_**	**X_31_**	**X_41_**	**X_51_**	**X_61_**	**X_71_**	**X_81_**	**X_91_**	**X_10,1_**
X_11_	-	(0,0)	(0.08,0)	(0,0.16)	(0.08,0)	(0,0.16)	(0,0.058)	(0.08,0)	(0,0)	(0,0.28)
X_21_	-	-	(0.08,0)	(0,0.16)	(0.08,0)	(0,0.16)	(0,0.058)	(0.08,0)	(0,0)	(0,0.28)
X_31_	-	-	-	(0,0.24)	(0,0)	(0,0.24)	(0,0.138)	(0,0)	(0,0.08)	(0,0.36)
X_41_	-	-	-	-	(0.24,0)	(0,0)	(0,0.102)	(0.2,0)	(0.16,0)	(0,0.12)
X_51_	-	-	-	-	-	(0,0.24)	(0,0.138)	(0,0)	(0,0.08)	(0,0.36)
X_61_	-	-	-	-	-	-	(0,0.102)	(0.2,0)	(0.16,0)	(0,0.12)
X_71_	-	-	-	-	-	-	-	(0.138,0)	(0.058,0)	(0.228,0)
X_81_	-	-	-	-	-	-	-	-	(0,0.8)	(0,0.36)
X_91_	-	-	-	-	-	-	-	-	-	(0,0.28)
X_10,1_	-	-	-	-	-	-	-	-	-	-
	**X_12_**	**X_22_**	**X_32_**	**X_42_**	**X_52_**	**X_62_**	**X_72_**	**X_82_**	**X_92_**	**X_10,2_**
X_12_	-	(0.06,0)	(0.06,0)	(0.06,0)	(0.12,0)	(0,0.03)	(0,0.039)	(0.06,0)	(0.13,0)	(0,0.061)
X_22_	-	-	(0,0)	(0,0)	(0.06,0)	(0,0.03)	(0,0.099)	(0,0)	(0.06,0)	(0,0.099)
X_32_	-	-	-	(0,0)	(0.06,0)	(0,0.03)	(0,0.099)	(0,0)	(0.06,0)	(0,0.099)
X_42_	-	-	-	-	(0.06,0)	(0,0.03)	(0,0.099)	(0,0)	(0.06,0)	(0,0.099)
X_52_	-	-	-	-	-	(0,0.09)	(0,0.159)	(0,0.06)	(0,0)	(0,0.059)
X_62_	-	-	-	-	-	-	(0,0.069)	(0.03,0)	(0.09,0)	(0.031,,0)
X_72_	-	-	-	-	-	-	-	(0.099,0)	(0.159,0)	(0.1,0)
X_82_	-	-	-	-	-	-	-	-	(0.06,0)	(0,0.001)
X_92_	-	-	-	-	-	-	-	-	-	(0,0.059)
X_10,2_	-	-	-	-	-	-	-	-	-	-
	**X_13_**	**X_23_**	**X_33_**	**X_43_**	**X_53_**	**X_63_**	**X_73_**	**X_83_**	**X_93_**	**X_10,3_**
X_13_	-	(0,0.04)	(0,0.04)	(0,0.12)	(0,0)	(0,0.03)	(0,0.148)	(0,0.13)	(0.04,0)	(0,0.193)
X_23_	-	-	(0,0)	(0,0.08)	(0.04,0)	(0,0.07)	(0,0.188)	(0,0.16)	(0.08,0)	(0,0.233)
X_33_	-	-	-	(0,0.08)	(0.04,0)	(0,0.07)	(0,0.188)	(0,0.16)	(0.08,0)	(0,0.233)
X_43_	-	-	-	-	(0.12,0)	(0,0.16)	(0.268,0)	(0,0.25)	(0.16,0)	(0,0.313)
X_53_	-	-	-	-	-	(0,0.03)	(0,0.148)	(0,0.13)	(0.03,0)	(0,0.193)
X_63_	-	-	-	-	-	-	(0,0.188)	(0,0.1)	(0.01,0)	(0,0.163)
X_73_	-	-	-	-	-	-	-	(0,0.018)	(0.108,0)	(0,0.045)
X_83_	-	-	-	-	-	-	-	-	(0.09,0)	(0,0.063)
X_93_	-	-	-	-	-	-	-	-	-	(0,0.153)
X_10,3_	-	-	-	-	-	-	-	-	-	-
	**X_14_**	**X_24_**	**X_34_**	**X_44_**	**X_54_**	**X_64_**	**X_74_**	**X_84_**	**X_94_**	**X_10,4_**
X_14_	-	(0.01,0)	(0.03,1)	(0,0)	(0.01,0)	(0.1,0)	(0,0.072)	(0,0)	(0.01,0)	(0,0.183)
X_24_	-	-	(0.04,0)	(0,0.01)	(0,0)	(0,0.09)	(0,0.082)	(0,0.01)	(0,0)	(0,0.193)
X_34_	-	-	-	(0,0.03)	(0,0.04)	(0,0)	(0,0.13)	(0,0.03)	(0,0.04)	(0,0.153)
X_44_	-	-	-	-	(0.01,0)	(0.1,0)	(0,0.072)	(0,0)	(0.01,0)	(0,0.183)
X_54_	-	-	-	-	-	(0,0.09)	(0,0.082)	(0,0.01)	(0,0)	(0,0.193)
X_64_	-	-	-	-	-	-	(0,0.172)	(0,0.1)	(0.09,0)	(0,0.283)
X_74_	-	-	-	-	-	-	-	(0.072,0)	(0.082,0)	(0,0.121)
X_84_	-	-	-	-	-	-	-	-	(0.01,0)	(0,0.183)
X_94_	-	-	-	-	-	-	-	-	-	(0,0.193)
X_10,4_	-	-	-	-	-	-	-	-	-	-
	**X_15_**	**X_25_**	**X_35_**	**X_45_**	**X_55_**	**X_65_**	**X_75_**	**X_85_**	**X_95_**	**X_10,5_**
X_15_	-	(0.12,0)	(0.12,0)	(0,0.16)	(0,0)	(0,0.04)	(0,0.02)	(0,0.06)	(0.12,0)	(0,0.232)
X_25_	-	-	(0,0)	(0,0.04)	(0,0.12)	(0,0.16)	(0,0.1)	(0,0.06)	(0,0)	(0,0.102)
X_35_	-	-	-	(0,0.04)	(0,0.12)	(0,0.16)	(0,0.1)	(0,0.06)	(0,0)	(0,0.102)
X_45_	-	-	-	-	(0.16,0)	(0.2,0)	(0.14,0)	(0.1,0)	(0.04,0)	(0.072,0)
X_55_	-	-	-	-	-	(0,0.04)	(0,0.02)	(0,0.06)	(0.12,0)	(0,0.232)
X_65_	-	-	-	-	-	-	(0,0.06)	(0,0.1)	(0.16,0)	(0,0.272)
X_75_	-	-	-	-	-	-	-	(0.04,0)	(0.1,0)	(0.212,0)
X_85_	-	-	-	-	-	-	-	-	(0.06,0)	(0.172,0)
X_95_	-	-	-	-	-	-	-	-	-	(0,0.112)
X_10,5_	-	-	-	-	-	-	-	-	-	-
	**X_16_**	**X_26_**	**X_36_**	**X_46_**	**X_56_**	**X_66_**	**X_76_**	**X_86_**	**X_96_**	**X_10,6_**
X_16_	-	(0.13,0)	(0.01,0)	(0,0.17)	(0.13,0)	(0,0.1)	(0,0.031)	(0,0.07)	(0.13,0)	(0,0.142)
X_26_	-	-	(0,0.12)	(0,0.04)	(0,0)	(0,0.03)	(0,0.099)	(0,0.06)	(0,0)	(0,0.012)
X_36_	-	-	-	(0,0.16)	(0.12,0)	(0,0.09)	(0,0.021)	(0,0.06)	(0.12,0)	(0,0.132)
X_46_	-	-	-	-	(0.04,0)	(0.07,0)	(0.139,0)	(0.1,0)	(0.04,0)	(0.028,0)
X_56_	-	-	-	-	-	(0,0.03)	(0,0.099)	(0,0.06)	(0,0)	(0,0.012)
X_66_	-	-	-	-	-	-	(0,0.069)	(0,0.03)	(0.03,0)	(0,0.042)
X_76_	-	-	-	-	-	-	-	(0,0.049)	(0.099,0)	(0,0.111)
X_86_	-	-	-	-	-	-	-	-	(0.06,0)	(0,0.072)
X_96_	-	-	-	-	-	-	-	-	-	(0,0.012)
X_10,6_	-	-	-	-	-	-	-	-	-	-
	**X_17_**	**X_27_**	**X_37_**	**X_47_**	**X_57_**	**X_67_**	**X_77_**	**X_87_**	**X_97_**	**X_10,7_**
X_17_	-	(0.03,0)	(0.04,0)	(0,0.07)	(0.03,0)	(0,0)	(0.031,0)	(0.03,0)	(0.03,0)	(0,0.042)
X_27_	-	-	(0,0.07)	(0,0.06)	(0,0)	(0,0.03)	(0,0.001)	(0,0)	(0,0.06)	(0,0.012)
X_37_	-	-	-	(0,0.11)	(0.07,0)	(0,0.04)	(0,0.071)	(0.07,0)	(0.01,0)	(0,0.082)
X_47_	-	-	-	-	(0.04,0)	(0.07,0)	(0.039,0)	(0.04,0)	(0.1,0)	(0.028,0)
X_57_	-	-	-	-	-	(0,0.03)	(0,0.001)	(0,0)	(0,0.06)	(0,0.012)
X_67_	-	-	-	-	-	-	(0.031,0)	(0.03,0)	(0.03,0)	(0,0.042)
X_77_	-	-	-	-	-	-	-	(0.001,0)	(0.061,0)	(0,0.011)
X_87_	-	-	-	-	-	-	-	-	(0,0.06)	(0,0.012)
X_97_	-	-	-	-	-	-	-	-	-	(0,0.072)
X_10,7_	-	-	-	-	-	-	-	-	-	-
	**X_18_**	**X_28_**	**X_38_**	**X_48_**	**X_58_**	**X_68_**	**X_78_**	**X_88_**	**X_98_**	**X_10,8_**
X_18_	-	(0.21,0)	(0.24,0)	(0,0.2)	(0.25,0)	(0,0.1)	(0.284,0)	(0,0.1)	(0.25,0)	(0,0.128)
X_28_	-	-	(0,0.03)	(0,0.01)	(0,0.04)	(0,0.11)	(0,0.174)	(0,0.11)	(0,0.04)	(0,0.082)
X_38_	-	-	-	(0,0.04)	(0.01,0)	(0,0.14)	(0.044,0)	(0,0.14)	(0.01,0)	(0,0.112)
X_48_	-	-	-	-	(0.05,0)	(0.1,0)	(0.084,0)	(0.1,0)	(0.05,0)	(0.072,0)
X_58_	-	-	-	-	-	(0,0.15)	(0,0.034)	(0,0.15)	(0,0)	(0,0.122)
X_68_	-	-	-	-	-	-	(0.184,0)	(0,0)	(0.15,0)	(0,0.028)
X_78_	-	-	-	-	-	-	-	(0,0.184)	(0.034,0)	(0,0.156)
X_88_	-	-	-	-	-	-	-	-	(0.15,0)	(0.028,0)
X_98_	-	-	-	-	-	-	-	-	-	(0,0.122)
X_10,8_	-	-	-	-	-	-	-	-	-	-
	**X_19_**	**X_29_**	**X_39_**	**X_49_**	**X_59_**	**X_69_**	**X_79_**	**X_89_**	**X_99_**	**X_10,9_**
X_19_	-	(0.06,0)	(0,0)	(0,0)	(0,0)	(0,0.1)	(0,0.15)	(0,0.03)	(0.06,0)	(0,0.039)
X_29_	-	-	(0,0.06)	(0,0.06)	(0,0.06)	(0,0.04)	(0,0.21)	(0,0.03)	(0,0)	(0,0.099)
X_39_	-	-	-	(0,0)	(0,0)	(0.0.1)	(0,0.15)	(0,0.03)	(0.06,0)	(0,0.039)
X_49_	-	-	-	-	(0,0)	(0,0.1)	(0.15,0)	(0.03,0)	(0.06,0)	(0.039,0)
X_59_	-	-	-	-	-	(0,0.1)	(0.15,0)	(0.03,0)	(0.06,0)	(0.039,0)
X_69_	-	-	-	-	-	-	(0.25,0)	(0.07,0)	(0.04,0)	(0.139,0)
X_79_	-	-	-	-	-	-	-	(0,0.18)	(0.21,0)	(0,0.111)
X_89_	-	-	-	-	-	-	-	-	(0.03,0)	(0.069,0)
X_99_	-	-	-	-	-	-	-	-	-	(0,0.099)
X_10,9_	-	-	-	-	-	-	-	-	-	-
	**X_1,10_**	**X_2,10_**	**X_3,10_**	**X_4,10_**	**X_5,10_**	**X_6,10_**	**X_7,10_**	**X_8,10_**	**X_9,10_**	**X_10,10_**
X_1,10_	-	(0,0)	(0,0)	(0,0.21)	(0,0.05)	(0,0.13)	(0,0.403)	(0,0.08)	(0,0)	(0,0.019)
X_2,10_	-	-	(0,0)	(0,0.21)	(0,0.05)	(0,0.13)	(0,0.403)	(0,0.08)	(0,0)	(0,0.019)
X_3,10_	-	-	-	(0,0.21)	(0,0.05)	(0,0.13)	(0,0.403)	(0,0.08)	(0,0)	(0,0.019)
X_4,10_	-	-	-	-	(0.16,0)	(0.08,0)	(0.613,0)	(0.13,0)	(0.21,0)	(0.191,0)
X_5,10_	-	-	-	-	-	(0,0.08)	(0.453,0)	(0.03,0)	(0.05,0)	(0.031,0)
X_6,10_	-	-	-	-	-	-	(0.533,0)	(0.05,0)	(0.13,0)	(0.111,0)
X_7,10_	-	-	-	-	-	-	-	(0,0.483)	(0.403,0)	(0,0.422)
X_8,10_	-	-	-	-	-	-	-	-	(0.08,0)	(0,0.061)
X_9,10_	-	-	-	-	-	-	-	-	-	(0,0.019)
X_10,10_	-	-	-	-	-	-	-	-	-	-
	**X_1,11_**	**X_2,11_**	**X_3,11_**	**X_4,11_**	**X_5,11_**	**X_6,11_**	**X_7,11_**	**X_8,11_**	**X_9,11_**	**X_10,11_**
X_1,11_	-	(0,0)	(0,0.07)	(0,0.14)	(0,0.07)	(0,0.14)	(0,0.001)	(0,0.06)	(0,0)	(0,0.012)
X_2,11_	-	-	(0,0.07)	(0,0.14)	(0,0.07)	(0,0.14)	(0,0.001)	(0,0.06)	(0,0)	(0,0.012)
X_3,11_	-	-	-	(0,0.21)	(0,0)	(0,0.21)	(0,0.071)	(0.01,0)	(0.07,0)	(0,0.082)
X_4,11_	-	-	-	-	(0.21,0)	(0,0)	(0.139,0)	(0.2,0)	(0.14,0)	(0,0.128)
X_5,11_	-	-	-	-	-	(0,0.21)	(0,0.071)	(0.01,0)	(0.07,0)	(0,0.082)
X_6,11_	-	-	-	-	-	-	(0.139,0)	(0.2,0)	(0.14,0)	(0,0.128)
X_7,11_	-	-	-	-	-	-	-	(0.061,0)	(0.001,0)	(0,0.011)
X_8,11_	-	-	-	-	-	-	-	-	(0.06,0)	(0,0.072)
X_9,11_	-	-	-	-	-	-	-	-	-	(0,0.012)
X_10,11_	-	-	-	-	-	-	-	-	-	-
	**X_1,12_**	**X_2,12_**	**X_3,12_**	**X_4,12_**	**X_5,12_**	**X_6,12_**	**X_7,12_**	**X_8,12_**	**X_9,12_**	**X_10,12_**
X_1,12_	-	(0,0.01)	(0.04,0)	(0,0.05)	(0,0.04)	(0,0.07)	(0,0.011)	(0,0.04)	(0.04,0)	(0,0.188)
X_2,12_	-	-	(0.03,0)	(0,0.04)	(0.05,0)	(0,0.06)	(0,0.001)	(0.05,0)	(0.03,0)	(0,0.178)
X_3,12_	-	-	-	(0,0.01)	(0,0.08)	(0,0.03)	(0,0.029)	(0,0.08)	(0,0)	(0,0.148)
X_4,12_	-	-	-	-	(0.09,0)	(0.02,0)	(0.039,0)	(0.09,0)	(0.01,0)	(0.138,0)
X_5,12_	-	-	-	-	-	(0,0.11)	(0,0.051)	(0,0)	(0.08,0)	(0,0,228)
X_6,12_	-	-	-	-	-	-	(0,0.059)	(0.11,0)	(0.03,0)	(0,0.118)
X_7,12_	-	-	-	-	-	-	-	(0.051,0)	(0.029,0)	(0,0.177)
X_8,12_	-	-	-	-	-	-	-	-	(0.08,0)	(0,0.228)
X_9,12_	-	-	-	-	-	-	-	-	-	(0,0.148)
X_10,12_	-	-	-	-	-	-	-	-	-	-

**Table 22 ijerph-17-03268-t022:** The concordance matrix.

	A1	A2	A3	A4	A5	A6	A7	A8	A9	A10
A1	-	(2.4,3.633,4.8)	(1.8,2.833,3.8)	(5.6,8.4,10)	(2.8,4.333,5.8)	(5.8,8.466,10.8)	(4.8,7.7,8.8)	(4.2,6.667,8.8)	(1.6,2.233,2.8)	(6.2,9.233,11.8)
A2	(3.8,5.6,7)	-	(4.6,6.866,8.8)	(6.2,9.233,11.8)	(4,6.1,7.8)	(6.2,9.233,11.8)	(6.2,9.233,11.8)	(5,7.633,9.8)	(4.8,7,8.8)	(6.2,9.233,11.8)
A3	(4.4,6.4,8)	(1.6,2.367,3)	-	(6.2,9.233,11.8)	(3.6,5.3,6.8)	(6.2,9.233,11.8)	(5.8,8.533,10.8)	(4,7.6,9.8)	(2.4,3.7,4.8)	(6.2,9.233,11.8)
A4	(0.6,0.833,1)	(0,0,0)	(0,0,0)	-	(0.6,0.8,1)	(3,4.033,5)	(1.8,2.433,3)	(1.4,2.3,3)	(0,0,0)	(3.4,4.8,6)
A5	(3.4,4.9,6)	(2.2,3.133,4)	(2.6,3.933,5)	(5.6,8.433,10.8)	-	(6.2,9.233,11.8)	(5.6,7.9,10)	(4.8,7.1,9)	(3.4,4.9,6)	(5.6,7.9,10)
A6	(0.4,0.767,1)	(0,0,0)	(0,0,0)	(3.2,5.2,6.8)	(0,0,0)	-	(3.8,5.567,7)	(2.4,3.834,5)	(0,0,0)	(4.8,7.067,9)
A7	(1,1.533,2)	(0,0,0)	(0.4,0.7,1)	(4.4,6.8,8.8)	(0.8,1.333,1.8)	(2.4,3.666,4.8)	-	(2.2,3.6,4.8)	(0,0,0)	(4.2,6.7,8.8)
A8	(2,2.566,3)	(1.2,1.6,2)	(1.2,1.633,2)	(4.8,6.933,8.8)	(1.4,2.133,2.8)	(3.8,5.399,6.8)	(4,5.633,7)	-	(1.4,1.733,2)	(4.6,6.933,8.8)
A9	(4.6,7,9)	(1.4,2.233,3)	(3.8,5.533,7)	(6.2,9.233,11.8)	(2.8,4.333,5.8)	(6.2,9.233,11.8)	(6.2,9.233,11.8)	(4.8,7.5,9.8)	-	(6.2,9.233,11.8)
A10	(0,0,0)	(0,0,0)	(0,0,0)	(2.8,4.433,5.8)	(0.8,1.333,1.8)	(1.4,2.166,2.8)	(2,2.533,3)	(1.6,2.3,3)	(0,0,0)	-

**Table 23 ijerph-17-03268-t023:** The disconcordance matrix.

	A1	A2	A3	A4	A5	A6	A7	A8	A9	A10
A1	-	1	1	0.286	1	0.625	0.077	0.8	1	0
A2	0	-	0.667	0	0.667	0	0	0.5	1	0
A3	0	1	-	0	1	0	0.234	0.438	1	0
A4	1	1	1	-	1	1	1	1	1	0.61
A5	0	1	0	0	-	0	1	0.2	1	0.108
A6	1	1	1	0	1	-	1	1	1	0.491
A7	1	1	1	0	0	0	-	0.286	1	0.54
A8	1	1	1	0	1	0	1	-	1	0.25
A9	0	0	0	0	0	0	0	0	-	0
A10	1	1	1	1	1	1	1	1	1	-

**Table 24 ijerph-17-03268-t024:** Boolean matrix B based on the minimum concordance level.

	A1	A2	A3	A4	A5	A6	A7	A8	A9	A10
A1	-	0	0	1	0	1	1	1	0	1
A2	1	-	1	1	1	1	1	1	1	1
A3	1	0	-	1	1	1	1	1	0	1
A4	0	0	0	-	0	1	0	0	0	1
A5	1	0	0	1	-	1	1	1	1	1
A6	0	0	0	0	0	-	1	0	0	1
A7	0	0	0	1	0	0	-	0	0	1
A8	0	0	0	1	0	1	1	-	0	1
A9	1	0	1	1	0	1	1	1	-	1
A10	0	0	0	0	0	0	0	0	0	-

**Table 25 ijerph-17-03268-t025:** Boolean matrix H based on the minimum discordance level.

	A1	A2	A3	A4	A5	A6	A7	A8	A9	A10
A1	-	0	0	1	0	0	1	0	0	1
A2	1	-	0	1	0	1	1	1	0	1
A3	1	0	-	1	0	1	1	1	0	1
A4	0	0	0	-	0	0	0	0	0	0
A5	1	0	1	1	-	1	0	1	0	1
A6	0	0	0	1	0	-	0	0	0	1
A7	0	0	0	1	1	1	-	1	0	1
A8	0	0	0	1	0	1	0	-	0	1
A9	1	1	1	1	1	1	1	1	-	1
A10	0	0	0	0	0	0	0	0	0	-

**Table 26 ijerph-17-03268-t026:** The global matrix Z.

	A1	A2	A3	A4	A5	A6	A7	A8	A9	A10
A1	-	0	0	1	0	0	1	0	0	1
A2	1	-	0	1	0	1	1	1	0	1
A3	1	0	-	1	0	1	1	1	0	1
A4	0	0	0	-	0	0	0	0	0	0
A5	1	0	0	1	-	1	0	1	0	1
A6	0	0	0	0	0	-	0	0	0	1
A7	0	0	0	1	0	0	-	0	0	1
A8	0	0	0	1	0	1	0	-	0	1
A9	1	0	1	1	0	1	1	1	-	1
A10	0	0	0	0	0	0	0	0	0	-

**Table 27 ijerph-17-03268-t027:** The results of fuzzy ELECTRE I method.

Alternative	Incomparable Alternative	Submissive Alternative	Ranking
A1	A6,A8	A4,A10	5
A2	A3,A5,A9	A1,A4,A6,A7,A8,A10	1
A3	A2	A1,A4,A5,A6,A7,A8	4
A4	A6,A10	-	9
A5	A2,A3,A7,A9	A1,A4,A6,A8,A10	1
A6	A1,A4,A7	A10	7
A7	A5,A6,A8	A4, A10	7
A8	A1,A7	A4,A6,A10	5
A9	A2,A5	A1,A3,A4,A6,A7,A8,A10	1
A10	A4	-	10

**Table 28 ijerph-17-03268-t028:** Comparing TOPSIS and ELECTRE I method results.

Rank	Rank in TOPSIS	Rank in ELECTRE
A1	5	5
A2	2	1
A3	3	4
A4	9	9
A5	4	1
A6	8	7
A7	7	7
A8	6	5
A9	1	1
A10	10	10

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
