# Peer review of "Green Supplier Selection Based on Green Practices Evaluated Using Fuzzy Approaches of TOPSIS and ELECTRE with a Case Study in a Chinese Internet Company"

_ijerph, 2020, doi:10.3390/ijerph17093268_

Round 1

Reviewer 1 Report

In this paper, author uses fuzzy TOPSIS and ELECTRE approach to choose green chain suppliers for Chinese internet company. This is an application-oriented paper, also get the appropriate results, but the paper has some problems need to be revised.

  1. This paper selected 12 criteria that should be considered when choosing green suppliers based on the literature review and the experts’ opinions. However, it is important to note whether the assessment value of the variable is appropriate for Chinese Internet companies. What does the author think?
  2. In the paper, the author uses the Fuzzy TOPSIS method, generally speaking, the Intuitionistic Fuzzy sets and Pythagorean sets are superior to traditional Fuzzy set of Fuzzy sets, then, using the Intuitionistic Fuzzy TOPSIS or Pythagorean fuzzy TOPSIS may get better results, the author is how to consider.
  3. In the fuzzy TOPSIS and ELECTRE approach, the author uses the distance measure. If the similarity is used, what result will be produced? How does the author think about it?
  4. In sensitivity analysis, what is the basis of table 19?
  5. In the paper, Fuzzy TOPSIS and ELECTRE approach are not that different. They are also not optimal methods, so the reliability of the results remains to be considered, especially in application articles.
  6. The structure of the article should be considered, e.g., section4, 4.1,5. In addition, the name section4 is not very appropriate。
  7. The last paragraph of the Introduction should be reorganized so that the logic is clearer.
  8. ELECTRE approach is not proposed by the author, but there are no references. In addition, consistent with the status of Fuzzy TOPSIS, why not Section3.
  9. The content of Section 2.2 is too confusing and illogical.
  10. In the introduction, the author mentioned that there would be limitations of paper in the conclusion. However, it was not found.
  11. Fuzzy TOPSIS and ELECTRE approach are both used, and the importance is the same, as can be seen from the title of the article. However, only fuzzy TOPSIS is mentioned in the conclusion, which is an unreasonable behavior.

In addition, there are still some minor issues that need to be fixed, as follows:

(1) Some of the descriptions in figure 1 are not clear, such as "Identify the Boolean matrices B and H -based"

(2) Where is the explanation in table 18.

(3) Please confirm that all references are in the same format.

Some recent works on MADM would be mentioned as follows:

Sun, C.; Li, S.Y.; Deng. Y. Determining Weights in Multi-Criteria Decision Making Based on Negation of Probability Distribution under Uncertain Environment. Mathematics. 2018, 8,191.

Fei, L.G.; Deng, Y. Multi-criteria decision making in Pythagorean fuzzy environment. Applied Intelligence. 2020, 50, 537—561.

Bian, T.; Zheng, H.Y; Yin, L.K; Deng, Y. Failure mode and effects analysis based on D numbers and TOPSIS. Quality and Reliability Engineering International. 2018, 34, 501—515.

In short, this work is applied to the actual work has reference significance. However, the problems in the paper cannot be ignored. I recommend to accept this paper after major revisions. 

Author Response

Paper ID:ijerph-772795

Dear Editors and Reviewers:

First of all, we sincerely thank you for your time and constructive comments on our manuscript entitled “Green Supplier Selection based on Green Practices Evaluated using Fuzzy Approaches of TOPSIS and ELECTRE with a Case Study in a Chinese Internet Company” (ID: ijerph-772795). Those comments are all valuable and very helpful for revising and improving our paper, as well as the important guiding significance to our researches. We have studied these comments carefully and have made correction which we hope meet with approval. Revised portion are marked in red in the paper. The main corrections in the paper and the point-to-point responses to the editor’s comments are as following:

Reviewer 1

In this paper, author uses fuzzy TOPSIS and ELECTRE approach to choose green chain suppliers for Chinese internet company. This is an application-oriented paper, also get the appropriate results, but the paper has some problems need to be revised.

Point 1:           This paper selected 12 criteria that should be considered when choosing green suppliers based on the literature review and the experts’ opinions. However, it is important to note whether the assessment value of the variable is appropriate for Chinese Internet companies. What does the author think?

Response 1: The judgment of different decision makers will not be exactly the same, so we invited three decision makers to make judgment to reduce the bias caused by subjective color. The standard estimate may vary from company to company. But for Chinese Internet companies, the direction is consistent. Therefore, it is necessary to combine the general with the special.

Point 2:           In the paper, the author uses the Fuzzy TOPSIS method, generally speaking, the Intuitionistic Fuzzy sets and Pythagorean sets are superior to traditional Fuzzy set of Fuzzy sets, then, using the Intuitionistic Fuzzy TOPSIS or Pythagorean fuzzy TOPSIS may get better results, the author is how to consider.

Response 2: We will use intuitionistic fuzzy TOPSIS or Pythagoras fuzzy TOPSIS in the subsequent research to further study this field. We have made relevant explanations in the future research directions in Section 5 from line 531 to line 540. Please check the revised version using the "Track Changes" function in Microsoft Word.

Point 3:In the fuzzy TOPSIS and ELECTRE approach, the author uses the distance measure. If the similarity is used, what result will be produced? How does the author think about it?

Response 3: Because there is a relationship between distance (d) and similarity (s), S (a, b) = 1-d (a, b), so this paper uses one of the variable measurements. The author thinks that the best supplier using the two is the same.

Point 4:           In sensitivity analysis, what is the basis of table 19?

Response 4: We have elaborated on the basis of Table 19 in the sensitivity analysis in more detail. See lines 414 to 420 of the revised manuscript. Please check the revised version using the "Track Changes" function in Microsoft Word.

Point 5:           In the paper, Fuzzy TOPSIS and ELECTRE approach are not that different. They are also not optimal methods, so the reliability of the results remains to be considered, especially in application articles.

Response 5: Thank you for your suggestion. In order to improve the reliability of the results, we conducted a sensitivity analysis on the results of the Fuzzy TOPSIS method, and compared with the results of the ELECTRE method to obtain the final optimal supplier selection.

Point 6:           The structure of the article should be considered, e.g., section4, 4.1,5. In addition, the name section4 is not very appropriate。

Response 6: We have corrected the chapter structure. And change the name of the section 4 to “Application of the proposed green supplier selection framework” on line 301. Section 4.2 may have been accidentally deleted during the modification process. Please check the revised version using the "Track Changes" function in Microsoft Word.

Point 7:           The last paragraph of the Introduction should be reorganized so that the logic is clearer.

Response 7: We have reorganized the last paragraph of the introduction to make the logic clearer. See lines 86 to 95 of the revised manuscript. Please check the revised version using the "Track Changes" function in Microsoft Word.

Point 8:           ELECTRE approach is not proposed by the author, but there are no references. In addition, consistent with the status of Fuzzy TOPSIS, why not Section3.

Response 8: We have added section 2.3 and section 3.3 to express the ELECTRE method,on line 162 and line 275 respectively. Please check the revised version using the "Track Changes" function in Microsoft Word.

Point 9:           The content of Section 2.2 is too confusing and illogical.

Response 9: We have adjusted the content of section 2.2 to make it more logical. Please check the revised version using the "Track Changes" function in Microsoft Word.

Point 10:In the introduction, the author mentioned that there would be limitations of paper in the conclusion. However, it was not found.

Response 10: We have added the limitation of this paper from line 531 to line 540 in section 5. Please check the revised version using the "Track Changes" function in Microsoft Word.

Point 11:Fuzzy TOPSIS and ELECTRE approach are both used, and the importance is the same, as can be seen from the title of the article. However, only fuzzy TOPSIS is mentioned in the conclusion, which is an unreasonable behavior.

Response 11: We have added information about the ELECTRE method to the conclusion. Please check the revised version using the "Track Changes" function in Microsoft Word.

Point 12:In addition, there are still some minor issues that need to be fixed, as follows:

(1) Some of the descriptions in figure 1 are not clear, such as "Identify the Boolean matrices B and H -based"

(2) Where is the explanation in table 18.

(3) Please confirm that all references are in the same format.

Response 12: (1) We have corrected the relevant unclear statements in Figure 1.

(2) We have added an explanation of Table 18 from line 414 to line 415.

(3) We have revised and confirmed that all references cited are in the same format.

Please check the revised version using the "Track Changes" function in Microsoft Word.

Point 13:Some recent works on MADM would be mentioned as follows:

Sun, C.; Li, S.Y.; Deng. Y. Determining Weights in Multi-Criteria Decision Making Based on Negation of Probability Distribution under Uncertain Environment. Mathematics. 2018, 8,191.

Fei, L.G.; Deng, Y. Multi-criteria decision making in Pythagorean fuzzy environment. Applied Intelligence. 2020, 50, 537—561.

Bian, T.; Zheng, H.Y; Yin, L.K; Deng, Y. Failure mode and effects analysis based on D numbers and TOPSIS. Quality and Reliability Engineering International. 2018, 34, 501—515.

Response 13: We have added the above recent work on MCDM in the manuscript. They are on line 58, line 62 and line 203. Please check the revised version using the "Track Changes" function in Microsoft Word.

Thank you again for your constructive comments that help us a lot to improve the paper.

Reviewer 2 Report

Review Report #   ijerph-772795

This study explores critical factors influencing green supplier selection by using multi-criterion decision making tools such as TOPSIS and ELECTRE subsidy. The manuscript can provide an interesting direction from the perspective of green supply chain practice and I recommend major revision. My revision suggestions are as follows:

Literature review section needs to be improved: for example, the following recent article deals with green supply chain management practice in the presence of government subsidy which is always important:

Is It a Strategic Move to Subsidized Consumers Instead of the Manufacturer?, IEEE Access 7, 169807-169824

Comparative analysis of government incentives and game structures on single and two-period green supply chain, Journal of Cleaner Production 235, 1371-1398

Game-Theoretic Analysis to Examine How Government Subsidy Policies Affect a Closed-Loop Supply Chain Decision, Applied Sciences 10 (1), 145

The authors used the ELECTRE, but there are many alternatives. Therefore, the authors need to discuss: What are the other feasible alternatives? What are the advantages of adopting these particular mechanisms over others in this case? How will this affect the results? More details should be furnished.

Some key parameters are not mentioned in numerical illustration. The rationale for the choice of the particular set of parameters should be explained with more details. What are the sensitivities of these parameters on the results?

The authors mentioned that “Comparing the TOPSIS method and the ELECTRE I method, there are the following similarities and differences”. The authors need to discuss the issue in detail.

Author Response

Response to Reviewer 2 Comments

Paper ID:ijerph-772795

Dear Editors and Reviewers:

First of all, we sincerely thank you for your time and constructive comments on our manuscript entitled “Green Supplier Selection based on Green Practices Evaluated using Fuzzy Approaches of TOPSIS and ELECTRE with a Case Study in a Chinese Internet Company” (ID: ijerph-772795). Those comments are all valuable and very helpful for revising and improving our paper, as well as the important guiding significance to our researches. We have studied these comments carefully and have made correction which we hope meet with approval. Revised portion are marked in blue in the paper. The main corrections in the paper and the point-to-point responses to the editor’s comments are as following:

Reviewer 2

This study explores critical factors influencing green supplier selection by using multi-criterion decision making tools such as TOPSIS and ELECTRE subsidy. The manuscript can provide an interesting direction from the perspective of green supply chain practice and I recommend major revision. My revision suggestions are as follows:

Point 1:Literature review section needs to be improved: for example, the following recent article deals with green supply chain management practice in the presence of government subsidy which is always important:

Is It a Strategic Move to Subsidized Consumers Instead of the Manufacturer?, IEEE Access 7, 169807-169824

Comparative analysis of government incentives and game structures on single and two-period green supply chain, Journal of Cleaner Production 235, 1371-1398

Game-Theoretic Analysis to Examine How Government Subsidy Policies Affect a Closed-Loop Supply Chain Decision, Applied Sciences 10 (1), 145

Response 1: We have cited the above recent artical. See lines 102 to 105 of the revised manuscript. Please check the revised version using the "Track Changes" function in Microsoft Word.

Point 2:The authors used the ELECTRE, but there are many alternatives. Therefore, the authors need to discuss: What are the other feasible alternatives? What are the advantages of adopting these particular mechanisms over others in this case? How will this affect the results? More details should be furnished.

Response 2: We have discussed alternative suppliers in the results discussion of Section 4. And other details have furnished in the discussion of Section 4 and the conclusion of Section 5. Please check the revised version using the "Track Changes" function in Microsoft Word.

Point 3:Some key parameters are not mentioned in numerical illustration. The rationale for the choice of the particular set of parameters should be explained with more details. What are the sensitivities of these parameters on the results?

Response 3: We have furnished more details of some key parameters in Section 4. See lines 314 to 343 of the revised manuscript. Please check the revised version using the "Track Changes" function in Microsoft Word.

Point 4:The authors mentioned that “Comparing the TOPSIS method and the ELECTRE I method, there are the following similarities and differences”. The authors need to discuss the issue in detail.

Response 4: We have enriched the comparison between TOPSIS method and ELECTRE I method from line 489 to line 497. Please check the revised version using the "Track Changes" function in Microsoft Word.

Thank you again for your constructive comments that help us a lot to improve the paper.

Reviewer 3 Report

  1. It would be good to better highlight why you are using the TOPSIS method in your paper. 
  2. Taking into consideration the limits of using the case of one Chinese company only would be fine. Can we make generalisations?
  3. Also, it would be good to tell the readers when the GSCM method started striking roots in management. 

Author Response

Response to Reviewer 3 Comments

Paper ID:ijerph-772795

Dear Editors and Reviewers:

First of all, we sincerely thank you for your time and constructive comments on our manuscript entitled “Green Supplier Selection based on Green Practices Evaluated using Fuzzy Approaches of TOPSIS and ELECTRE with a Case Study in a Chinese Internet Company” (ID: ijerph-772795). Those comments are all valuable and very helpful for revising and improving our paper, as well as the important guiding significance to our researches. We have studied these comments carefully and have made correction which we hope meet with approval. Revised portion are marked in blue in the paper. The main corrections in the paper and the point-to-point responses to the editor’s comments are as following:

Reviewer 3

This study explores critical factors influencing green supplier selection by using multi-criterion decision making tools such as TOPSIS and ELECTRE subsidy. The manuscript can provide an interesting direction from the perspective of green supply chain practice and I recommend major revision. My revision suggestions are as follows:

Point 1: It would be good to better highlight why you are using the TOPSIS method in your paper. 

 Response 1: We have explained why use TOPSIS method from line 55 to line 65. Please check the revised version using the "Track Changes" function in Microsoft Word.

Point 2: Taking into consideration the limits of using the case of one Chinese company only would be fine. Can we make generalisations?

Response 2: We have explained this in Section 5. We propose a framework for supplier selection for corporate reference and apply it to an Internet company in China. In future research, we will continue to innovate the framework to improve its reliability.

Point 3: Also, it would be good to tell the readers when the GSCM method started striking roots in management.

Response 3: We have enriched the relevant content about the application of the GSCM method in management in the introduction. See lines 36 to 43 of the revised manuscript. Please check the revised version using the "Track Changes" function in Microsoft Word.

Thank you again for your constructive comments that help us a lot to improve the paper.

Round 2

Reviewer 1 Report

The problem has been fixed. Work can be accepted

Reviewer 2 Report

I have no further comments. The paper can be accepted.